# Causes and consequences of experimental variation in *Nicotiana benthamiana* transient expression

Sophia N. Tang [1,2,3], Matthew J. Szarzanowicz[1,2,4], Amy Lanctot[1,2,4], Sasilada Sirirungruang[1,2,4], Liam D. Kirkpatrick[1,2,4], Krista Drako [1,2,3], Simon Alamos [1,2,4], Lyurui Cheng [1,2,5], Lucas M. Waldburger[1,2,6,7], Shuying Liu[1,2], Lena Huang [1,2], Sami Kazaz [1,2], Emine Akyuz Turumtay[1], Edward Baidoo[1], Aymerick Eudes[1,2], Mitchell G. Thompson [1,2] ✉ & Patrick M. Shih [1,2,4,8] ✉

Infiltration of *Agrobacterium tumefaciens* into *Nicotiana benthamiana* has become a foundational technique in plant biology, enabling efficient delivery of transgenes *in planta* with technical ease, robust signal, and relatively high throughput. Despite transient expression's prevalence in disciplines such as synthetic biology, little work has been done to describe and address the variability inherent in this system, a concern for experiments that rely on highly quantitative readouts. In a comprehensive analysis of *N. benthamiana* agroinfiltration experiments, we model sources of variability that affect transient expression. Our findings emphasize the need to validate normalization methods under the specific conditions of each study, as distinct normalization schemes do not always reduce variation either within or between experiments. Using a dataset of 1915 plants collected over three years, we develop a model of variation in *N. benthamiana* transient expression, using power analysis to determine the number of individual plants required for a given effect size. Drawing on our longitudinal data, these findings inform practical guidelines for minimizing variability through strategic experimental design and power analysis, providing a foundation for more robust and reproducible use of *N. benthamiana* in quantitative plant biology and synthetic biology applications.

*Nicotiana benthamiana* is a workhorse of plant molecular biology due to the high efficiency of transient *Agrobacterium*-mediated transformation, enabling researchers to assess gene function and observe phenotypes within days, as opposed to the months it typically takes for stable transformations[1–3]. *N. benthamiana*'s capacity for transient expression precipitated its use to study diverse aspects of cell and molecular biology[4], including virus-induced gene silencing[5], subcellular protein localization[6], biosynthetic pathway discovery and engineering[7–11], and gene regulation[12–16]. Given the lengthy time required to generate stable transformants and the technical challenges inherent in protoplast transformation[17,18], there are few viable *in planta* alternatives to *N. benthamiana*

[1]Joint BioEnergy Institute, Emeryville, CA, USA. [2]Environmental Genomics and Systems Biology Division, Lawrence Berkeley National Laboratory, Berkeley, CA, USA. [3]Department of Molecular and Cell Biology, University of California, Berkeley, CA, USA. [4]Department of Plant and Microbial Biology, University of California, Berkeley, CA, USA. [5]Department of Nutritional Sciences and Toxicology, University of California, Berkeley, CA, USA. [6]Department of Bioengineering, University of California, Berkeley, CA, USA. [7]Biological Systems & Engineering Division, Lawrence Berkeley National Laboratory, Berkeley, CA, USA. [8]Innovative Genomics Institute, Berkeley, CA, USA. ✉e-mail: mthompson@lbl.gov; pmshih@berkeley.edu

transient expression, leading to its primacy as the model of plant synthetic biology[19,20].

Despite its widespread use, N. benthamiana lacks substantial efforts to assess error and reproducibility, like many other systems routinely leveraged in synthetic biology[21]. Nor is there a standardized way to design transient assays, and as a result, experimental design varies from publication to publication[13,22-26]. These factors obstruct our ability to model and predict variation in this model system, which is crucial for effective experimental design, as has been demonstrated in the closely related Nicotiana tabacum[27]. Limited quantitative understanding of biology has routinely impeded genetic engineers in developing predictable and reliable bioproducts[28], in comparison to the precision associated with traditional engineering disciplines[29]. For plant synthetic biology to truly be an engineering discipline, we require robust evaluation of variability and its effects on our design goals.

The variability of transient expression in N. benthamiana in highly powered and longitudinal experiments has never been rigorously evaluated. Without robust statistics, researchers risk designing underpowered experiments and overlooking subtle effects. While there have been attempts to determine transgene expression variability in stable plant lines[30,31], it is unclear whether these findings translate to transient expression. Transient expression variability in N. benthamiana has been primarily studied using luciferase reporters, but these studies are plagued by small sample size, lack of biological replicates, and the high noise inherent to the luciferase assay[32-34]. Most methods used to reduce variability in transient expression experiments have relied on using a second reporter (hereafter referred to as a normalizer) for ratiometric normalization[13,22-24,35,36]. In this approach, every tested experimental construct is co-delivered with the same constitutively expressed normalizer, under the assumption that any variation in the experimental construct's expression that tracks with variation in the normalizer's expression must not be due to the sequence of the reporters but rather other variables. Although this approach is widely used, whether normalizing transgene expression actually reduces and controls for variation has not been systematically examined.

Here, we use fluorescence reporter assays to systematically determine the sources of variation of transient expression in N. benthamiana leaves. Using a large sample size (>1900 plants) across multiple years of independent experiments, we comprehensively capture transgene expression variation observed between plants and between independent experiments in order to model sources of error. We also complete a systematic comparison of methods to normalize transgene expression in N. benthamiana and demonstrate whether and to what extent these methods reduce variability. Finally, we suggest best practices to mitigate variation through purposeful experimental design and statistical power analysis.

## Results

### Categorizing the sources of transient expression variation

To determine sources of variability in N. benthamiana transient expression, we analyzed previously published data wherein the same GFP reporter (vector 1 of Supplementary Table 1) was agroinfiltrated in 15 independent experimental replicates using A. tumefaciens GV3101::pMP90 (hereafter GV3101) (Fig. 1A and Supplementary Fig. 1)[37]. The reporter strain was infiltrated distal to the petiole of the fourth and fifth leaves from the top (leaves T4 and T5) of N. benthamiana, and three days post-infiltration, four disks were collected from each infiltrated leaf. All plants germinated at the same time and which experienced identical growing conditions and care are referred to as belonging to the same batch. In this dataset, we observed as much as a fourfold difference between experimental replicates with the lowest (teal, 2022.07.25) and highest (orange, 2023.05.30) mean fluorescence (Fig. 1B, C). Using a mixed-effects model, we attributed

nearly all of the variation to fluctuations in batch-level mean GFP (23.8%) and GFP standard deviation (15.6%); plant-level mean GFP (9.9%) and GFP standard deviation (28.2%) within a given batch; and disc-level GFP standard deviation (22.3%) within a given plant (Fig. 1D). Though the precise percent contributions of these components varies between experimental replicates (Supplementary Fig. 2), they are nonetheless all major sources of variation that should be accounted for in experimental design. The underlying factors of these sources of variation may be batch effects across the fifteen experimental replicates' plants, non-homogeneous growing conditions, and varying leaf cellular ages (Fig. 1E)[38,39].

Since the experimental conditions across these 15 replicates were identical, we separately investigated other potential contributors to variation that were controlled for in that experiment. In other independent experiments, we assessed the effects of 96-well water volume, plant age, leaf infiltration site, and collection time (both time of day and time elapsed between collection and measurement). We found that water volume and plant age are negatively correlated with fluorescence (Supplementary Figs. S3 and S4), infiltration site significantly affects transgene expression in a leaf- and strain-dependent manner (Supplementary Fig. 5), and collection time has no effect (Supplementary Figs. S6 and S7). Minor changes to water volume (e.g., evaporation, pipetting error) may have contributed to residual error in the data from Fig. 1B. However, all plants in this study were 4 weeks old at the time of infiltration and infiltrated distal to the petiole, so no variation can be attributed to differences in age or infiltration site. With these additional experiments, we have both accounted for the majority of the variation in our model and now ruled out other potential sources.

To demonstrate that these findings are generalizable beyond fluorescent proteins (FPs), we also quantified variation for two different metabolic pathways to produce betalain (3 experimental replicates) and 2-pyrone-4,6-dicarboxylic acid (PDC) (4 experimental replicates)[40,41]. The three enzymes of the betalain pathway were co-infiltrated on three separate T-DNAs or infiltrated as one T-DNA separated by self-cleaving T2A peptides (RUBY reporter)[42]. The absorbance of one of the three betalain experimental replicates was significantly different from the other two for both methods of delivery, and the greatest fold change between two replicates was about twofold for co-infiltration and about fourfold for T2A, on par with the maximal difference observed for the GFP reporter (Supplementary Fig. 8 and Fig. 1B–D). For the PDC experimental replicates, yields were not significantly different after Bonferroni correction, but there was a ~20% difference between the highest and lowest yields (Supplementary Fig. 8). Multi-step metabolic pathway yields are subject to enzyme subcellular localization, substrate diffusion, and the pathway's underlying dynamics, which may mask or exacerbate the effects of variable transgene expression. There are considerable variations in yields, specific to each pathway, and the trends observed in FP fluorescence likely translate to the variability of each component transgene in a metabolic pathway.

### Multiple methods of co-delivering a normalizing transgene can decrease expression variability

Normalizing the expression level of a transgene of interest to that of another independent transgene is thought to control for any sources of variability independent of the tested sequences, since they should impact the expression of the two transgenes equally. This practice should theoretically reduce variation both within and between experiments. However, there are many different conceivable ways the two transgenes could be delivered in N. benthamiana, and indeed, normalization schemes differ between publications[13,22-26].

We systematically compared normalization schemes for transient expression using FP reporters, whose gene expression is correlated to an easily measured phenotype, fluorescence.

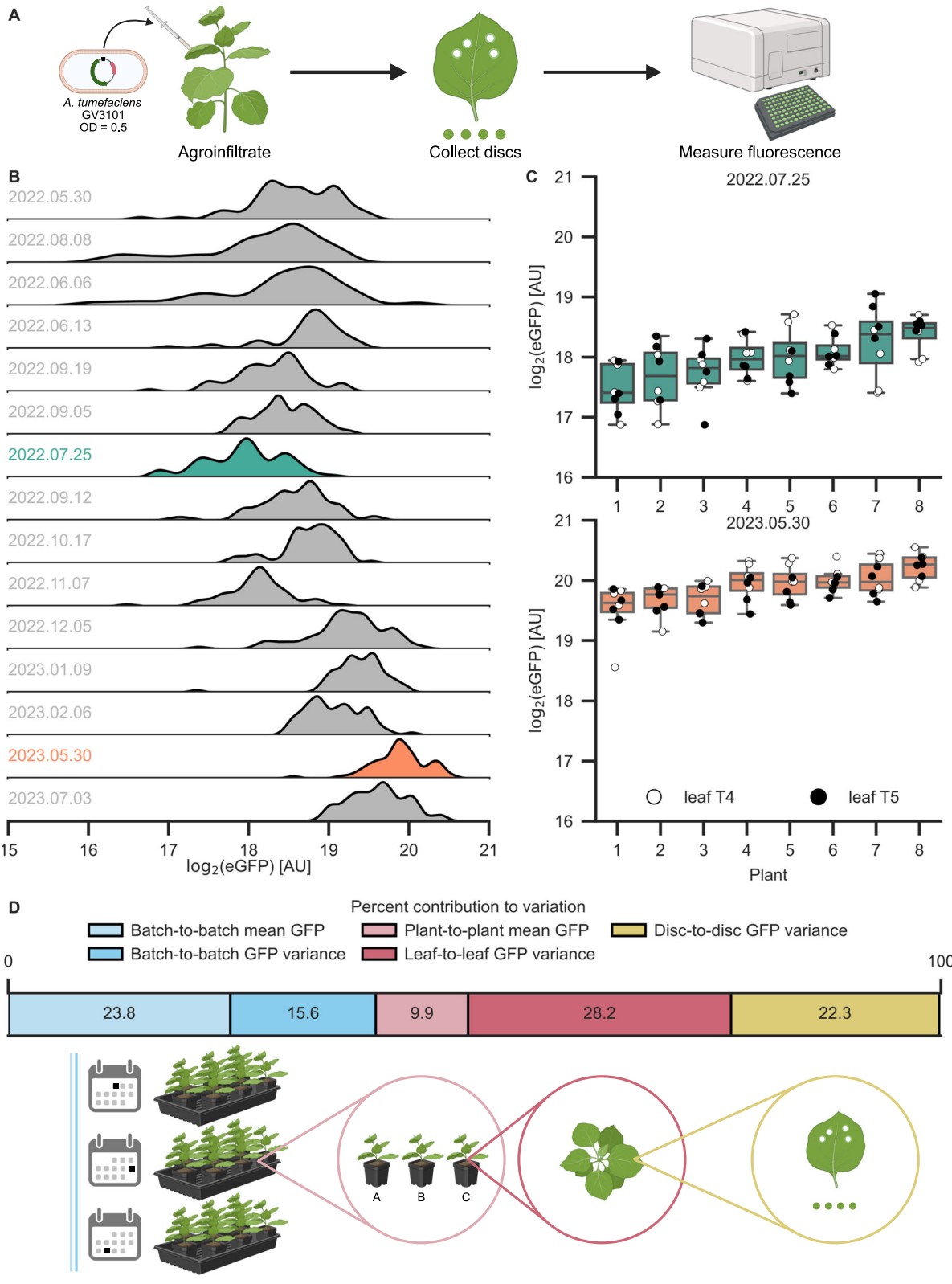

The normalized readout of the assay is the ratio of eGFP and mCherry fluorescence measured in a leaf disc. Designating FPs as the reporter or normalizer is arbitrary. The two expression cassettes were co-delivered in three different ways. First, we co-infiltrated a mix of two different strains of *A. tumefaciens*, each with a binary vector containing one FP (schemes 3–7). Second, we infiltrated single *A. tumefaciens* strains carrying binary vectors with a T-DNA containing both cassettes in every possible orientation, both relative to one another and to the left and right borders (schemes 8–15). Third, we infiltrated a "BiBi" strain[43], wherein a single strain of *A. tumefaciens* carries two binary vectors, each with a unique FP gene and ori (schemes 16–19). All FP genes are driven by the medium strength PCM2 constitutive promoter derived from the *A. thaliana* HTR5 histone gene (AT4G40040) and are terminated by the *A. thaliana* Ubq3 terminator (Supplementary Table 1 and Supplementary Fig. 1)[14]. These 17 methods of co-delivery were compared to

**Fig. 1 | Longitudinal assessment of transient expression identifies sources of variability in *N. benthamiana* fluorescence assays. A** Schematic of the transient fluorescence assay. *N. benthamiana* is agroinfiltrated with an *A. tumefaciens* strain at an OD of 0.5. The binary vector contains, from left to right border, CaMV35S2:nptII, and PCM2:eGFP:At_Ubq3 oriented divergently in the T-DNA. Disks are collected from leaves 3 days post-infiltration and measured in a plate reader. **B** Kernel density estimate plots of GFP fluorescence for 15 experimental replicates, $n = 64$ disks from 8 separate plants, except for 2022.06.06 and 2022.06.13, which are $n = 40$. Dates of sample collection and measurement are written YYYY.MM.DD. **C** All plants in experimental replicates with the lowest (2022.07.25) and highest (2023.05.30) mean raw fluorescence. Boxes show the median and interquartile range (IQR), and whiskers show the minima and maxima, excluding outliers (beyond 1.5 IQR). Scatterplots show T4 disks in white and T5 disks in black. In both experimental replicates, four disks were collected from both infiltrated leaves of 8 plants, for a total of $n = 64$ disks per experimental replicate. **D** Percent contributions to observed variability as calculated from a mixed-effects model and illustrations of sources of variability. Created in BioRender. Tang, S. (2025) https://BioRender.com/i2ej92v. Source data for this figure is available in the Source Data file.

delivering eGFP (scheme 1) or mCherry (scheme 2) alone in two independent experimental replicates.

Co-delivery methods have obvious ramifications for basal transgene expression levels that are consistent between both experimental replicates (Supplementary Fig. 9). For both co-infiltrations and BiBis, the low copy number origin, pSa, consistently produced much lower fluorescence compared to the higher copy number origins, pVS1 or BBR1 (Fig. 2A)[44]. For the stacked, two-cassette T-DNAs, despite sharing the same pVS1 origin and backbone, these schemes yielded up to eightfold difference in fluorescence depending on the cassettes' relative orientations (Fig. 2A and Supplementary Fig. 10). Convergent, tandem, and divergent cassettes produced the strongest, intermediate, and weakest expression, respectively (Supplementary Fig. 10). The expression of one gene is affected by the orientation of the other gene, even when its own position and orientation is maintained, as with schemes 8 and 10 for eGFP or schemes 9 and 11 for mCherry (Supplementary Fig. 10). The formation of a double terminator with the tOcs spacer in some cassettes may be partly responsible for increased expression, as has been shown with other binary vectors[45], but reversing tOcs does not consistently increase fluorescence from a doubly terminated cassette over the corresponding singly terminated cassette (Supplementary Fig. 10). These results suggest that gene position relative to the left and right borders and, most importantly, relative orientation strongly affect expression in multi-gene T-DNAs.

Variability, as measured by coefficient of variation (CV) from all leaf disks, decreases for all normalization schemes (Supplementary Fig. 11). Furthermore, all schemes have a lower CV for reporter/normalizer than for reporter (Supplementary Fig. 12). The more appropriate comparison, however, is whether a scheme's reporter/normalizer CV is lower than the corresponding unnormalized control's reporter CV. Most schemes produce an eGFP/mCherry CV that is significantly less than scheme 1's eGFP CV following a Bonferroni correction (Fig. 2B), but only scheme 3 (co-infiltration of two pVS1 binary vectors) meets this condition when mCherry is treated as the reporter (Fig. 2C). Such a test is conservative, and it should be noted that there is at least one scheme per co-delivery method that reduces CV by >50% compared to the unnormalized control, whether CV is calculated per plant or pooling all disks together (Fig. 2B, C and Supplementary Fig. 11). All co-delivery methods can offer reductions in variability, thereby increasing the statistical power of transient expression-based experiments.

## Inoculum densities of co-infiltrated strains affect signal strength but not variability

Because the decrease in CV was the greatest and most statistically significant for co-infiltration of two pVS1 binary vectors (scheme 3), we used this scheme to explore other possible experimental variables for further reductions in variability. Additionally, co-infiltration is highly modular, and multiple T-DNAs can efficiently be delivered to each plant cell[43,46]. Since *A. tumefaciens* strains in *N. benthamiana* leaves can antagonize or synergize with one another depending on their densities[43], we next determined whether the $OD_{600}$ of each strain in the co-infiltration affects the resulting CV. Each strain was added to the infiltration mix at an $OD_{600}$ of 0, 0.01, 0.1, 0.5, or 1 and combined with the other strain, also added at the same densities, for a total of 24 possible combinations.

Total $OD_{600}$ of the co-infiltrated mix and the $OD_{600}$ of the individual strains do not greatly or predictably affect variability, but the inoculum density of a strain dictates its total transgene expression levels. There is no discernible pattern in eGFP CV with respect to the $OD_{600}$ of the co-infiltrated strains (Fig. 3A), and while all CVs for the ratio of eGFP/mCherry are comparatively much lower (Fig. 3B), there is no one combination of ODs that is significantly superior to all others. The same trends are apparent when mCherry is treated as the reporter and eGFP as the normalizer (Fig. S13). For raw fluorescence, signal begins to saturate beyond an $OD_{600}$ of 0.5, and when one strain is held at a given $OD_{600}$ while the $OD_{600}$ of another competing strain increases, fluorescence from the fixed strain decreases (Fig. 3C, D). The primary concern when selecting $OD_{600}$, then, should be the desired signal strength. Of the densities tested, 0.1 results in a strong, measurable signal and, in agreement with previous findings, does not saturate the transgene expression capacity of the plant cell[43,46].

## Promoter choice determines the utility of normalization

Given that the choice of promoter(s) in transient transformations affects not only signal strength but also normalization outcomes[14], we more thoroughly explored additional promoter combinations. All transgenes in this study have hitherto been driven by a medium-strength promoter, PCM2. From a library of low, medium, and high-strength constitutive promoters characterized by Zhou et al. (annotated as "PCL", "PCM", and "PCH", respectively), we selected low- and high-strength promoters, PCL2 and PCH5, to test alongside PCM2. eGFP and mCherry were driven by this same set of three promoters ("same promoter set"). The eGFP binary vectors were co-infiltrated with the mCherry binary vectors in every possible combination, and all binary vectors were also infiltrated alone for a total of 15 unique infiltrations.

There is no discernible pattern in eGFP CV for any of the infiltrations (Fig. 4A), but for eGFP/mCherry, identical promoters driving the two FP genes yield the lowest CVs (Fig. 4B). Outside of these pairs, normalization does not provide as large reductions in variations, if at all. In fact, for three nonidentical promoter pairs, their eGFP/mCherry CV is greater than their own eGFP CV and, for one pair, also greater than the eGFP CV of its corresponding unnormalized control (Fig. 4A, B). Similar observations are true when treating mCherry as the reporter and eGFP as the normalizer (Supplementary Fig. 14). Additionally, fluorescence from a given strain diminishes the stronger a competing strain's FP promoter is, though this trend is weaker for mCherry than it is for GFP (Fig. 4C, D).

We then sought to clarify whether the large decrease in CV when using identical promoters is due to the similar promoter strengths or to other variables, such as shared trans factors affecting mRNA abundance. To do so, we generated another set of three binary vectors with mCherry driven by PCL1, PCM1, and PCH4 ("different promoter set"), which are the promoters in the library closest in expression to the three already tested. As with the first promoter combination experiment, the existing eGFP binary vectors were co-infiltrated with the new

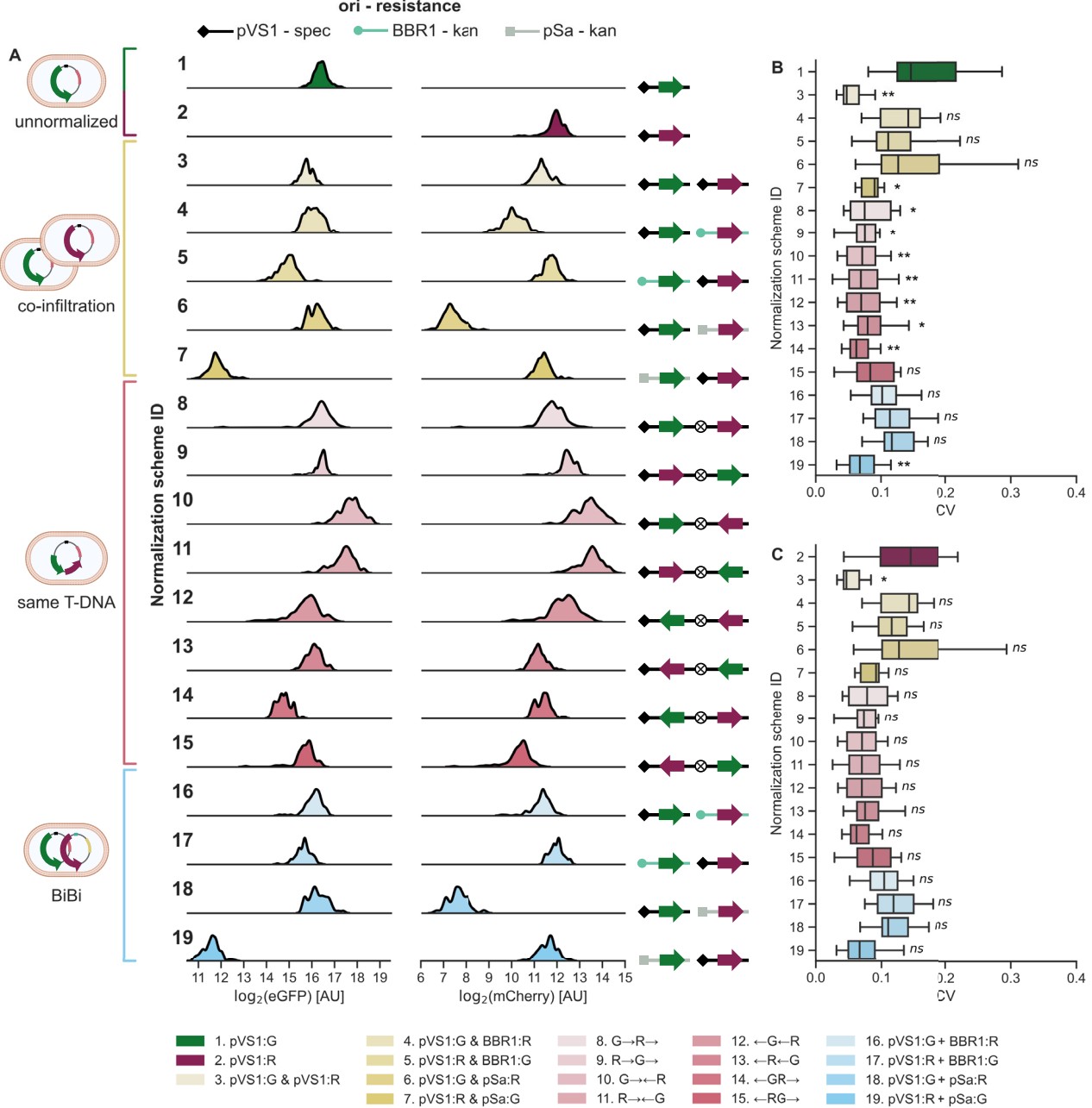

**Fig. 2 | Comparison of three discrete methods to deliver normalizing reporters.** **A** Left, categories of delivery methods: unnormalized (green or magenta), co-infiltration (yellow), same T-DNA (pink), and BiBi (blue). Center, kernel density estimation plots of eGFP and mCherry fluorescence, $n = 96$ leaf disks per row. Right, cartoons showing the binary vector origin of replication, resistance marker, and orientations of FP expression cassettes in the T-DNA. All binary vector cartoons are read from left to right: ori, left border, T-DNA, right border. Origins of replication are pVS1 (diamond, black), BBR1 (circle, teal), and pSa (square, gray). Circles enclosing an X represent tOcs, a 722 bp spacer in between the two expression cassettes. **B** Plant coefficients of variation (CV), as calculated from the 8 disks per plant, when eGFP is treated as the reporter. All values are eGFP/mCherry CV except for scheme 1, which is GFP CV. **C** Plant CVs when mCherry is treated as the reporter. All values are mCherry/eGFP CV except for scheme 2, which is mCherry CV. Normalization scheme IDs match across all subpanels. Boxes show the median and IQR,

and whiskers show the minima and maxima, excluding outliers (beyond 1.5 IQR). For **B**, **C**, an independent, one-tailed Welch's $t$-test and a Bonferroni correction were conducted to determine whether the reporter/normalizer CV of a scheme is significantly less than the reporter CV for the corresponding unnormalized scheme. Two experimental replicates of six plants are shown in (**B**, **C**), for a total of $n = 12$ plants. Asterisks indicate $p$-values: *<0.05, **<0.01. Total OD infiltrated in all schemes is 0.5. ODs of co-infiltrated strains are 0.25 each. In the legend, "&" indicates co-infiltration, arrows indicate the direction of an expression cassette, "+" indicates BiBi, and GFP is abbreviated to "G" and mCherry to "R". Leaves T4 and T5 of six four-week-old *N. benthamiana* plants were infiltrated in both experimental replicates, and four disks were collected from each infiltrated leaf. Created in BioRender. Tang, S. (2025) https://BioRender.com/0573uvz. Source data for this figure is available in the Source Data file.

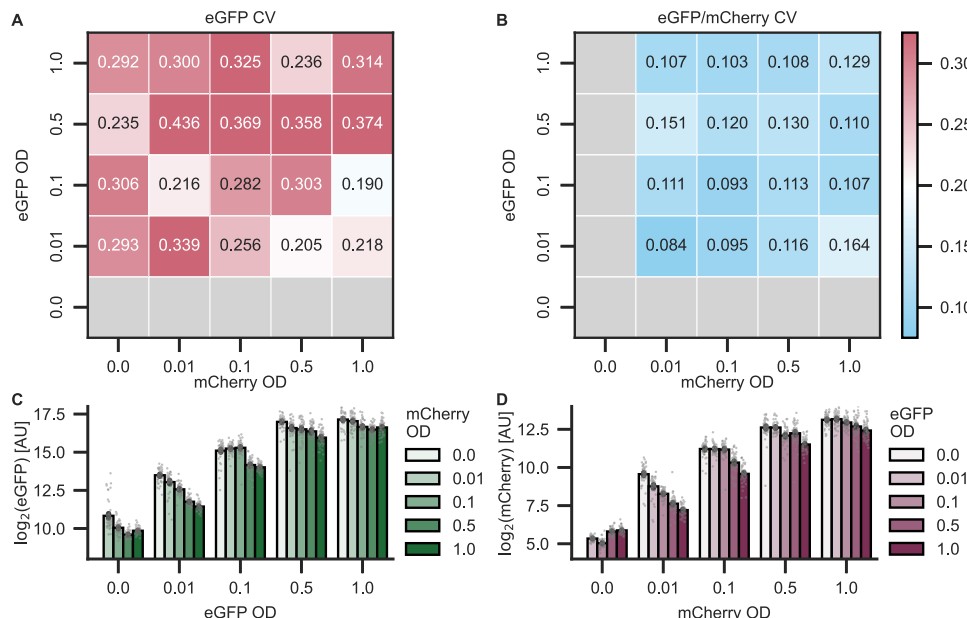

**Fig. 3 | OD$_{600}$ of two co-infiltrated strains affects transgene expression strength but not variability.** Matrix of all OD$_{600}$ combinations' CV of **A** eGFP fluorescence and **B** ratio of eGFP/mCherry. Raw fluorescence signal from **C** eGFP and **D** mCherry for all OD$_{600}$ combinations. Darker hues indicate increasing OD$_{600}$ of the competing strain, which carries the other FP. Error bars show the standard error. Each condition was infiltrated into leaves T4 and T5 of six plants. Four disks were collected from each leaf for a total of $n = 48$ leaf disks. Source data for this figure is available in the Source Data file.

mCherry binary vectors in every possible combination, and all six binary vectors were also infiltrated alone.

With these promoters, both the CVs of eGFP and of eGFP/mCherry follow no apparent pattern (Fig. 4E, F), but the negative trend between fluorescence signals of the two competing strains holds true, although PCH4 fits poorly in the trend (Fig. 4G, H). Several promoter pairs have higher eGFP/mCherry CVs than their own eGFP CVs and/or the eGFP CV of the unnormalized control (Fig. 4E, F), and vice versa when treating mCherry as the reporter (Supplementary Fig. 14). Similarity in promoter strength, then, does not guarantee lower CVs upon normalization. Depending on the promoters used to normalize, normalization may even worsen the assay's statistical power. Since the co-delivery comparison and the OD optimization experiments were conducted using PCM2 for every expression cassette, we had fortuitously picked the best case scenario, identical promoters. Had we chosen nonidentical promoters, all methods of co-delivery might not have decreased CV compared to the unnormalized controls. When possible, identical promoters should be used.

### Normalization does not yield reproducibility of absolute quantification

Normalization can serve two distinct purposes: (1) decreasing within-experiment variation to improve detection of small effect sizes, or (2) decreasing between-experiment variation to allow for comparison across independent experiments. Mitigating within-experiment variation does not necessarily have an effect on between-experiment variation. To explore whether normalization makes comparisons between experimental replicates or to historical data more valid, we co-infiltrated GFP driven by PCL2 (which yielded the lowest average normalized CVs when driving both FPs (Fig. 4 and Supplementary Fig. 11)) alone and with all six mCherry binary vectors thus far generated in six independent experimental replicates.

Normalization makes experimental replicates more similar to one another for most, but not all, promoter pairs (Fig. 5A, B and Supplementary Fig. 15). We performed one-sample Kolmogorov-Smirnov tests for every condition to determine the likelihood that a sample distribution (an individual experimental replicate) is drawn from a reference distribution (all experimental replicates, pooled). The cumulative density functions (CDFs) show that sample distributions nearly all become more comparable to the reference distribution with normalization, i.e., the maximum vertical distance between the distributions decreases (Fig. 5A, B and Supplementary Fig. 15). Replicate-to-replicate, the PCL2/PCM2 promoter pair produces the most consistent distribution of GFP/mCherry values, far more reproducible than its raw eGFP fluorescence (Fig. 5A, B). The sole exception is the PCL2/PCM1 promoter pair, whose GFP/mCherry CDFs are actually less comparable to the reference distribution than its GFP CDFs are (Fig. 5A, B); there is no singular GFP/mCherry value that normalization with this promoter pair causes the data to reliably converge upon. However, this pair and PCL2/PCL2 were the only promoter pairs for which experimental replicate CVs, representing within-experiment variation, were regularly lower than the unnormalized control's (Fig. 5C). Additionally, PCL2/PCL2 produces the most consistent experimental replicate CVs–it alone has an interquartile range of experimental replicate CVs narrower than the unnormalized control's (Fig. 5C). Decreasing within-experiment variation, then, is not equivalent to decreasing between-experiment variation. Normalization schemes must be designed with the desired outcome in mind (minimizing within- and/or between-experiment variation) and then validated.

### Developing a generalizable power analysis framework of *N. benthamiana* transient expression

To evaluate whether normalization meaningfully reduces the number of plants required to detect a given effect size, we developed a model informed by a large dataset of *N. benthamiana* transient expression experiments. This dataset comprises 1813 plants infiltrated with a GFP reporter over nearly three years, spanning experiments conducted by multiple researchers and diverse conditions, including different inoculum densities and *Agrobacterium* strains (see Supplementary Method 2 for details). Because this dataset accounts for historical variation across diverse experiments, the simulations trained with it will give conservative estimates of variance. Individual, well-controlled experiments may yield lower variations, but since the precise

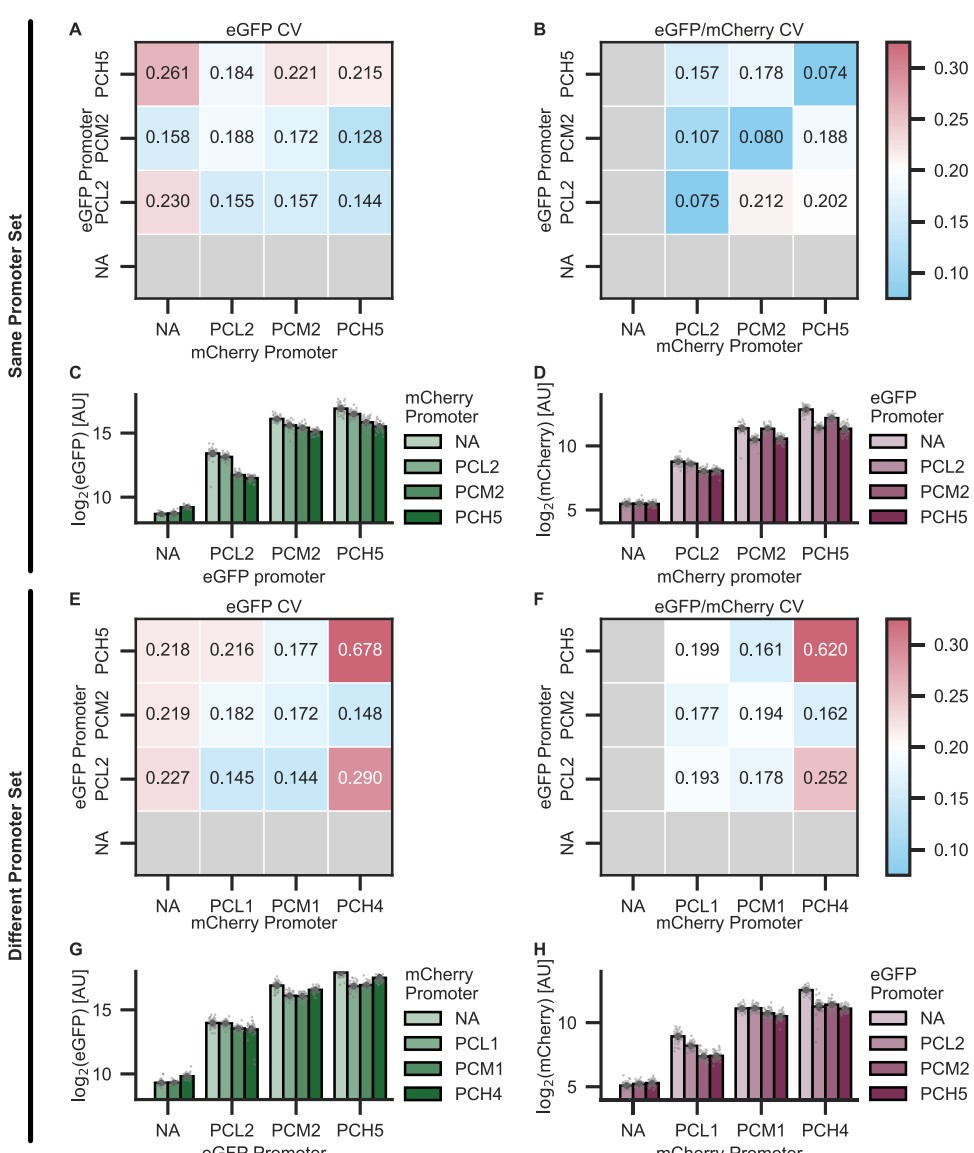

**Fig. 4 | Promoters of co-infiltrated transgenes determine the efficacy of normalization.** eGFP and mCherry were driven by PCL2, PCM2, or PCH5 for a total of six unique binary vectors. Matrix of all promoter combinations' CV of **A** eGFP fluorescence and **B** ratio of eGFP/mCherry. Log$_2$ raw fluorescence from **C** eGFP and **D** mCherry for all promoter combinations. Darker hues indicate a stronger promoter of the competing strain. Error bars show the standard error. Matrix of all promoter combinations' CV of **E** eGFP fluorescence and **F** ratio of eGFP/mCherry. Log$_2$ raw fluorescence signal from **G** eGFP and **H** mCherry for all promoter combinations when instead the mCherry binary vectors are driven by PCL1, PCM1, or PCH4. Darker hues indicate a stronger promoter of the competing strain. Error bars show the standard error. Conditions from the "Same Promoter Set" (**A**–**D**) and "Different Promoter Set" (**E**–**H**) experiments were infiltrated into leaves T4 and T5 of six plants, and four disks were collected from each leaf for a total of $n = 48$ leaf disks. The experiments were performed independently on separate days. Source data for this figure is available in the Source Data file.

variability cannot be known beforehand, erring on the side of excessive rather than insufficient statistical power is preferable. We calculated the CV for each plant infiltrated with one of two *Agrobacterium* strains: GV3101 and the hypervirulent EHA105, which displays both elevated transgene expression and variance compared to GV3101 (Fig. 6A). For batches of plants for which at least 30 plants were used, we visualized the plant CVs in a CDF (Fig. 6B, solid gray line). With these per-plant and per-experiment CVs, we developed a Monte Carlo simulation of assay variability, which closely recapitulates the observed distribution of real plant CVs across the training dataset (Fig. 6B dotted black line; Wasserstein distance = 0.023, mean ECDF difference = 0.05).

We then simulated how variability propagates through an experiment on the per-plant and per-experiment levels. First, a batch CV is drawn from the empirical distribution shown in Fig. 6A by randomly generating a percentile and selecting the corresponding CV, which represents the average CV of plants from a hypothetical experimental replicate. Around this batch-level value, we simulate CVs of individual plants by incorporating additional noise that approximates within-experiment heterogeneity between plants. For each simulated plant, eGFP fluorescence data for eight leaf disks are generated based on the plant CV and a fixed, construct-specific mean expression. Two conditions are compared with an independent, two-tailed Student's *t*-test at varying effect size differences and sample sizes (Fig. 6C). For each comparison, 1000 independent experimental replicates are simulated and used to calculate the probability of a successful comparison.

This approach enables us to estimate the minimum number of plants needed to reliably (>95% of simulations) and significantly ($p < 0.05$) distinguish between two constructs. From these calculations

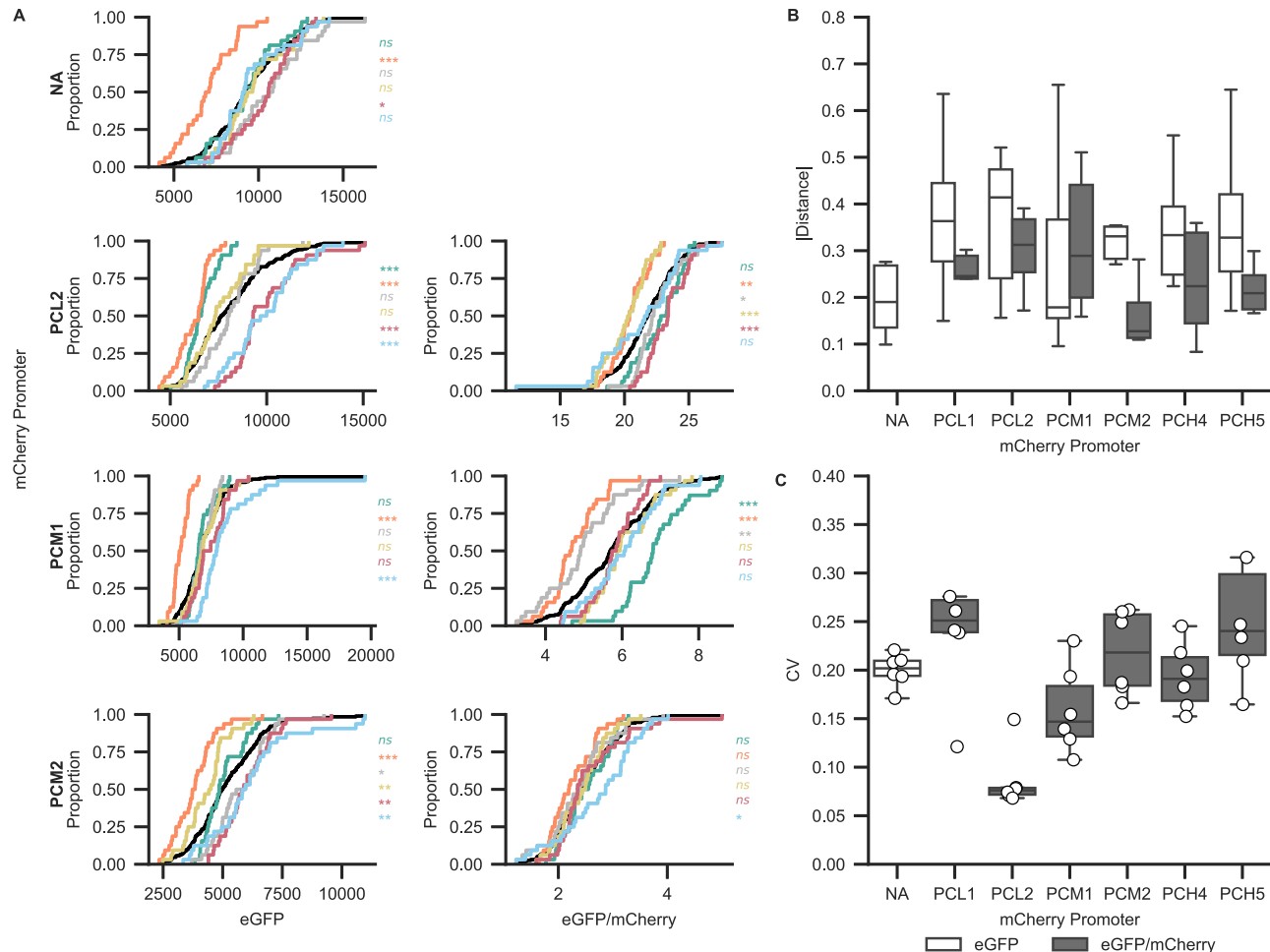

**Fig. 5 | Promoter choice affects variation between experimental replicates.**
**A** CDF of PCL2:eGFP alone or normalized by mCherry driven by PCL2, PCM1, or PCM2. Left: eGFP, right: eGFP/mCherry. Each experimental replicate is a unique color. The black line is the CDF for the pooled data of all six experimental replicates. The p-values of one-sample Kolmogorov-Smirnov tests appear to the right of each CDF, colored by experimental replicate. Asterisks indicate p-values: `*`<0.05, `**`<0.01, `***`<0.001, and ns = not significant. **B** Absolute value of every experimental replicate's D, the greatest vertical distance between the CDF of a given experimental replicate

(color) and the pooled CDF (black) from (**A**). White boxplots indicate D values for eGFP CDFs and gray boxplots for eGFP/mCherry CDFs. **C** CVs of the six experimental replicates for each condition. As in **B**, white boxplots indicate eGFP and gray eGFP/mCherry CDFs. Boxes show the median and IQR, and whiskers show the minima and maxima, excluding outliers (beyond 1.5 IQR). Every condition was infiltrated into leaves T4 and T5 of four plants, and four disks were collected from each leaf in all six experimental replicates, for a total of n = 192 leaf disks. Source data for this figure is available in the Source Data file.

of plants needed for given effect sizes, we fit an exponential decay curve assuming fixed CV for three situations: unnormalized EHA105, unnormalized GV3101, and optimally normalized GV3101 (lowest CV achieved in this publication) (Fig. 6D). The smallest detectable effect size using 50 plants for unnormalized EHA105 was 13.7%, compared to 10.7% for unnormalized GV3101 and 10.1% for GV3101 under optimal normalization (Fig. 6D). While larger sample sizes can resolve even smaller differences, practical constraints limit the number of plants that can be included in an experiment, but relatively small effect sizes can be accurately distinguished with a small number of plants. Based on the Monte Carlo simulations, with n = 3, the minimum reliably detectable effect size for unnormalized GV3101 is ~40%. Normalization is most beneficial in experiments where small effect sizes must be resolved using limited sample sizes. However, since normalization may also reduce statistical power in some cases (Supplementary Fig. 16), any normalization strategy should be validated empirically prior to implementation.

## Discussion

Here, we systematically measure expression variability in *N. benthamiana* agroinfiltration. The inherent noise in the transient

expression of FPs is considerable (as much as fourfold difference in mean expression across experimental replicates) but mitigable through proper experimental design (Fig. 1B). Using a mixed-effects model, we categorize nearly all observed variation in our dataset (Fig. 1D). We find that multiple co-delivery methods for dual reporters can decrease variation via ratiometric normalization, and of the examined methods, co-infiltration best reduces variation, approximately halving the CV compared to no normalization (Fig. 2). Neither inoculum density nor promoter strength in co-infiltrations predictably affect expression variation, but normalization efficacy is highly sensitive to promoter choice (Figs. 3 and 4). Some promoter pairs may, in fact, increase variation or make experimental replicates less comparable, and using identical promoters is the surest and most effective way to reduce CV within experiments (Figs. 4 and 5). Normalization should not be done arbitrarily but rather validated on a case-by-case basis for the co-delivery method and experimental design used, as this is absolutely necessary for results that are reproducible and robust to potential sources of variability.

Though unambiguously beneficial, the use of constitutive, identical promoters is also particularly limiting given the popularity of *N. benthamiana* to characterize the effects of promoter identity and

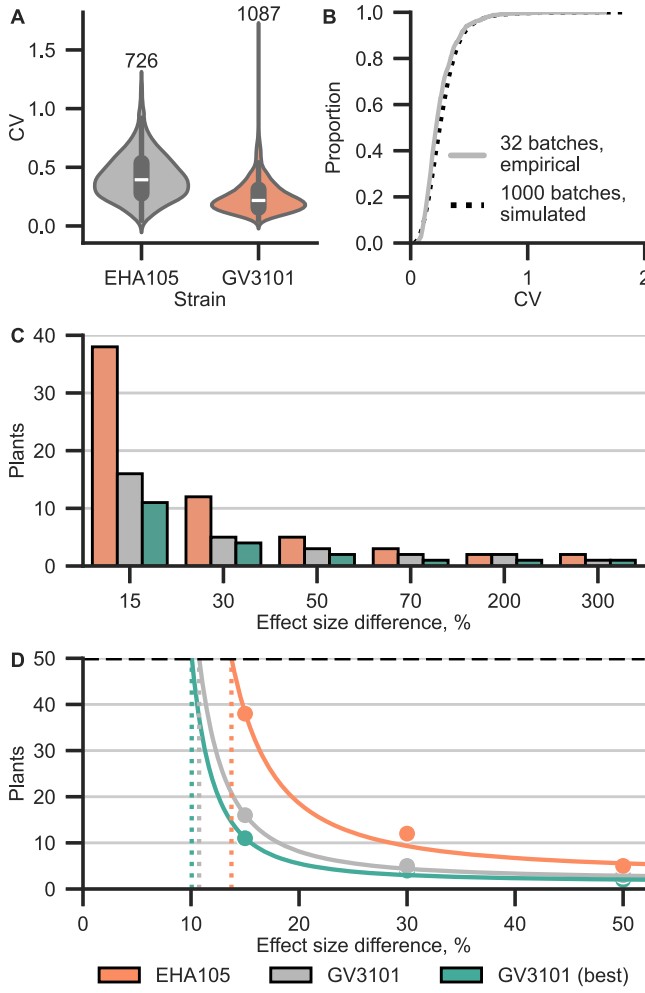

**Fig. 6 | Modeling variation of transgene expression in 4-week-old *N. benthamiana* plants. A** Per-plant CV of unnormalized GFP calculated from data compiled across many years and independent experiments. CV for a plant is calculated using all disks derived from that plant, regardless of the leaf. For EHA105, $n = 726$, and for GV3101, $n = 1087$. Violins show the distribution of all data. Boxes show the median and IQR, and whiskers show the minima and maxima, excluding outliers (beyond 1.5 IQR). **B** CDF of per-plant, unnormalized eGFP CVs. Solid gray, empirical data from 32 batches of at least 30 plants. Dotted black, Monte Carlo simulated data. **C** Minimum number of simulated plants needed to detect a given effect size with the CV of unnormalized EHA105, unnormalized GV3101, or optimally normalized GV3101, assuming 95% accuracy and statistical significance ($p < 0.05$) for a two-tailed Student's *t*-test. **D** Exponential regression fit to the Monte Carlo simulated data in (**C**), which are shown as points. Dashed horizontal line indicates the arbitrary 50 plant cap, and dotted vertical lines indicate the smallest detectable effect size with 50 plants. Orange, EHA105. Gray, unnormalized GV3101. Teal, optimally normalized GV3101. Source data for this figure is available in the Source Data file.

architecture on transcription. This problem is undoubtedly compounded in more complex experimental designs. To increase predictability and standardization of normalization methods, multi-lab efforts to quantify variation using diverse methods will be necessary. More robust transcriptomic datasets of *N. benthamiana* expressing diverse transgenes in distinct host strains could discourage reliance on a single, unvalidated normalizer. Additionally, multi-factor analysis of binary vector construction and expression cassette positioning could elucidate the extent to which these variables affect both expression and variability.

There is little published analysis of how the design of multi-gene T-DNA structure affects the transient expression of those genes[47]. We find that convergent cassettes are far more highly expressed than tandem or divergent cassettes. Since *Agrobacteria*'s VirD2 virulence protein covalently binds the right border[48], it is thought that transgenes proximal to the right border would be protected from exonucleolytic degradation and thus more likely to remain intact inside the plant cell, but this hypothesis is not fully supported by our data (Fig. 2A and Supplementary Fig. 10). Clearly, T-DNA architecture affects transgene expression strength and variation, and a high-throughput library approach would be ideal to address the multiplicity of orientation and position combinations in multi-gene binary vectors. Determining these rules of expression and assessing them in both stable and transient systems requires further, data-intensive work.

Given the lack of large, curated datasets of transient gene expression in *N. benthamiana*, it is likely that many published *N. benthamiana* experiments chose sample sizes arbitrarily based on unvalidated assumptions. Our model of the statistical power required to determine different effect sizes with high confidence shows that in cases where the expected effect size is large, it is easy to achieve the necessary sample size. For example, for effect sizes >50%, fewer than three plants are required using the GV3101 strain. For effect sizes <20%, many plants are required, to the extent that the number of plants needed may be infeasible for single experiments (e.g., 50 plants are needed to show a 13.7% effect using EHA105). The majority of synthetic biology applications for transient expression, such as bioproduction or circuit reconstruction, require strong effects, which call for small numbers of plants. However, these results could have strong implications for subtle phenotypes, like the transformation efficiency of various *Agrobacterium* strains or mutants[37,49], or inducible gene expression systems[47,50,51].

While the data in this work required many hundreds of plants and multiple years to collect, we were nonetheless only able to capture variation in *N. benthamiana* transient expression from a rather narrow perspective. All of our data are from fluorescent reporters from 4-week-old plants, with data collected three days after infiltration. We demonstrated that *Agrobacterium* strain, promoter choice, and T-DNA design all impact variation, but many more strains, plasmids, and expression cassette designs remain untested. There are other variables affecting *Agrobacterium*-mediated transformation that we did not address, such as the binary vector origin of replication or T-DNA length. Furthermore, as novel *Agrobacterium*-independent technologies to introduce nucleic acids and proteins into plant cells mature[52,53], their variability should also be estimated. It is critically important to establish whether our values of variance are broadly applicable, and future work should prioritize cross-lab validation.

Even in this simple context, variation in *N. benthamiana* transient expression depends strongly on a host of inputs, and attempts to normalize and mitigate this variation must be individually validated. As complexity increases in synthetic biology experimental designs, this unpredictability in variation and in confidence will likely be compounded. Our results show that the choice and validation of normalization methods and data collection are critical for *N. benthamiana* to be a reliable platform for applications such as reconstituting complex metabolic pathways and designing large genetic circuits. As we have thoroughly demonstrated, careful, purposeful experimental design and interpretation are paramount to ensure robust and reproducible results when using this essential technique for synthetic plant biology.

## Methods

### Media, chemicals, and culture conditions
Routine bacterial cultures were grown in Luria–Bertani (LB) Miller medium (BD Biosciences, USA). *E. coli* was grown at 37 °C, while *A. tumefaciens* was grown at 30 °C. Cultures were supplemented with rifampicin (100 µg/mL), kanamycin (50 µg/L, Sigma Aldrich, USA), gentamicin (30 µg/L, Fisher Scientific, USA), or spectinomycin (100 µg/L, Sigma Aldrich, USA), when indicated.

## Bacterial strain and plasmid construction

All bacterial strains and plasmids used in this work are listed in Supplementary Table 1. All strains and plasmids created in this work are viewable through the public instance of the Joint BioEnergy Institute (JBEI) registry: https://public-registry.jbei.org/folders/929. All strains and plasmids created in this work can be requested from the strain archivist at JBEI with a signed material transfer agreement. Plasmids were assembled by Gibson assembly using standard protocols (New England Biolabs). Plasmids were routinely isolated using the QIAprep spin miniprep kit (Qiagen), and all primers were purchased from Integrated DNA Technologies (IDT). Plasmid sequences were verified using whole-plasmid sequencing (Primordium Labs). *Agrobacterium* was routinely transformed by electroporation as described previously using a 1-mm cuvette and a 2.4-kV, 25-μF, 200-Ω pulse[54].

## *N. benthamiana* growth conditions

Wild-type *N. benthamiana* (LAB accession) plants were obtained from the in-house seed bank at the JBEI. All seedlings and plants were grown at 25 °C in 60% humidity under long-day conditions (16 h of light, 8 h of darkness) of 150 μmol m$^{-2}$ s$^{-1}$ photosynthetically active radiation (PAR; wavelength: 400–700 nm).

As many pots as needed of Sungro Sunshine mix #4 (aggregate plus) were wetted by running excess tap water from above and allowing it to drain through. The soil was then topped with a layer of topsoil that was thoroughly wetted with a spray bottle. Using a spatula, a pinch of seeds was sprinkled evenly over the soil. Ideally, 50–100 good seedlings per pot germinate. If seeds are sown too densely, the seedlings are smaller, and having more sparse, healthy seedlings results in superior plant quality at later stages. All seedling pots were placed in a flat with 1 L tapwater and covered with a hood, vents closed, in the growth room. If algal or cyanobacterial contamination grows in the water, reduce the volume of water added to the tray.

After one week, flats (as many as needed) were filled with 18 3.11" × 3.11" × 2.25" Traditional Inserts pots (Greenhouse Megastore). These pots were filled to the top with Sungro Sunshine mix #4 (aggregate plus), supplemented with 1.5 Tbsp Osmocote (14–14–14) pellets per 4 L soil, and then 3 L of tap water was added to the flat. The soil was allowed to soak up the water for 2–3 h, and then any excess was drained off and the pots broken apart. Small holes were made in each pot's soil, and the germinated seedlings were transplanted into individual pots. The root of the seedling must not be damaged during transplantation. The block of soil from a seedling pot can be removed and placed on its side to facilitate removing seedlings from the soil without causing damage. Flats were covered with a hood, vents closed, in the growth room. After one week, all the vents were opened to allow the hood to slowly equilibrate to the ambient conditions in the growth room. Five days later, the hoods were removed. Two days later, each flat was watered with tap water supplemented with Peter's Professional (20–20–20) at 1 tsp per 4 L water. After five days, the pots were rearranged into a checkerboard pattern within the flats to maximize the amount of space per plant, decreasing the density from 18 plants per flat to 9 plants per flat. Each flat was also watered with 1 L of water. Three days later, plants were infiltrated. A highly detailed protocol of *N. benthamiana* growth and care is provided in Supplementary Method 1.

## Agroinfiltration of *N. benthamiana*

Generated binary vectors were transformed into *A. tumefaciens* strain GV3101 via electroporation[54]. BiBi strains were generated by growing a liquid culture of single transformants to saturation, pelleting and washing thrice with ice cold 10% glycerol, resuspending in 100 μL water, and electroporating the second binary vector into the cells. Selected transformants were inoculated in liquid media with appropriate selection the night before the experiment. *A. tumefaciens* strains were grown until OD$_{600}$ between 0.8 and 1.2, centrifuged for 10 min at

4000×*g*, and resuspended in infiltration buffer (10 mM MgCl2, 10 mM MES, and 200 μM acetosyringone, pH 5.6) to achieve the desired OD$_{600}$. Cultures were induced for 1 h at room temperature on a rocking shaker. Leaves T4 and T5 of 4-week-old *N. benthamiana* plants were syringe-infiltrated with the *A. tumefaciens* suspensions. After infiltration, *N. benthamiana* plants were maintained in the same growth conditions as described above with 1 L tap water per tray. Three days post infiltration, four 6-mm leaf disks per infiltrated leaf were hole punched. The leaf disks were placed abaxial side up on 350 μL of water in black, 360-μL 96-well Costar Assay Plates with clear flat bottoms (Corning). eGFP (Ex.λ = 488 nm, Em.λ = 520 nm) and mCherry (Ex.λ = 587 nm, Em.λ = 615 nm) fluorescence were measured using a Synergy 4 microplate reader (BioTek). Gain was set at 100, and read height at 10.5 mm.

## RUBY extraction and quantification

Leaves T4 and T5 of 4-week *N. benthamiana* were co-infiltrated with equal amounts of three strains (carrying vectors 2–4 from Supplementary Table 1) on one side of the leaf and infiltrated with one strain (carrying vector 5 from Supplementary Table 1) on the other. Total OD$_{600}$ for both delivery methods was 0.5, and the infiltrated spots were not touching. Leaf tissues infiltrated with RUBY constructs were excised with a razor blade from the leaf 5 dpi, frozen in liquid nitrogen, and lyophilized for 2 d. Tissues were then weighed to record dry weights and homogenized with metal beads in a PowerLyzer at 1000×*g* for 2 min. 80 μL of 20% methanol per mg of dry weight was added to each sample. Samples were then centrifuged at 15,000 rcf for 5 min, and 100 μL of the supernatant was diluted with an additional 500 μL of water. Absorbances at λ = 538 nm of 300 μL of the diluted extracts in black, 360-μL 96 well Costar Assay Plates with clear flat bottoms (Corning) were measured in a Synergy 4 microplate reader (Bio-tek).

## PDC extraction and quantification

The entirety of leaves T4 and T5 of 4-week old *N. benthamiana* were infiltrated with 0.2 OD$_{600}$ of each of the five enzymes in the PDC pathway (total OD$_{600}$ = 1). At 5 dpi, both leaves were frozen in liquid nitrogen together and lyophilized for 2 d. Leaves were then ball-milled and extracted with 80% (*v/v*) methanol-water as solvent as previously described[40]. Metabolites were analyzed using an HPLC-ESI-TOF-MS as previously described and quantified with a 6-point calibration curve of PDC standard[40,41]. The monoisotopic *m/z* (negative ionization) of deprotonated PDC is 182.99351. A fermentation-monitoring HPX-87H column with 8% cross-linkage (150-mm length, 7.8-mm inside diameter, and 9-μm particle size; Bio-Rad, Richmond, CA) was used to separate metabolites with an Agilent Technologies 1100 Series HPLC system. Sample injection volumes of 10 μL. The sample tray and column compartments were set to 4 and 50 °C, respectively. Metabolites were eluted isocratically with a mobile-phase composition of 0.1% formic acid in water at a flow rate of 0.5 mL/min. The HPLC system was coupled to an Agilent Technologies 6210 series time-of-flight mass spectrometer (for LC-TOF MS) via a MassHunter workstation (Agilent Technologies, CA). Drying and nebulizing gases were 13 L/min and 30 lb/in$^2$, respectively, and drying-gas temperature was 330 °C. ESI was conducted in the negative ion mode, and the capillary voltage was −3500 V.

## Statistics and reproducibility

No statistical method was used to predetermine sample size, as the work itself is an attempt to characterize the variability inherent to the system, without making any assumptions about the necessary sample size to detect a given effect size. To that end, arbitrarily large sample sizes were chosen; all experimental conditions include at least six plants, two leaves per plant, four disks per leaf, for a total of *n* = 48 disks, with the exception of the six independent experiment replicates in Fig. 5, which each include four plants, for a total of *n* = 32 disks. Leaf

disks were excluded if they were infiltrated with an eGFP-containing binary vector, and the resulting green fluorescence was less than 1000. Similarly, disks were excluded if they were infiltrated with an mCherry-containing binary vector, and the resulting red fluorescence was <100. These fluorescence values are within the range of an uninfiltrated leaf, so these disks are assumed to be erroneously taken from uninfiltrated tissue. Plants were selected at random from different flats (locations) in the plant growth room, as opposed to picking adjacent plants within the room, which might introduce biases into the quality of plants in each group (and therefore into the data), as the conditions within the growth room are not perfectly uniform. The investigators were not blinded to allocation during experiments and outcome assessment.

### Reporting summary

Further information on research design is available in the Nature Portfolio Reporting Summary linked to this article.

## Data availability

All raw data related to this study are publicly available on GitHub (https://github.com/shih-lab/benthi_variation/tree/main/01-data) and Zenodo (DOI: 10.5281/zenodo.18004005[55]). Prior data reused for this work is available in the Source Data file. All plasmid sequences have been deposited to NCBI and are available under GenBank accession numbers PX927304-PX927337 (available at https://www.ncbi.nlm.nih.gov/genbank/)— see Supplementary Table 1 for individual accession codes. Source data are provided with this paper.

## Code availability

Mixed effects model and Monte Carlo simulations were run using R (v4.2.0) and the following packages: tidyverse (v2.0.0)[56], transport (v0.14.6), car (v3.1.3), lme4 (v1.1.31)[57], and performance (v0.13.0)[58]. All other analyses and figures were generated with Python (v3.11.4) and the following packages: jupyterlab (v4.0.3)[59], pandas (v2.3.3), numpy (v2.3.3)[60], seaborn (v0.13.2), matplotlib (v3.10.6), scipy (v1.16.1)[61], and statsmodels (v0.14.5). All code related to this study is publicly available on GitHub (github.com/shih-lab/benthi_variation/tree/main/02-code) and Zenodo (DOI: 10.5281/zenodo.18004005[55]).

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

## Acknowledgements

We would like to thank Dr. Tyler Backman and Dr. Hector Garcia-Martin for their invaluable advice on statistics, and to all members of the Shih lab for their helpful discussions. We also thank Konami for facilitating the discussion of expression cassette orientation via Dance Dance Revolution artwork. BioRender was used to generate Fig. 1 (https://BioRender.com/i2ej92v) and 2 (https://BioRender.com/0573uvz), as well as Supplementary Figs. S1 (https://biorender.com/u99vtsf), S5 (https://biorender.com/4fpcf4p), S9 (https://BioRender.com/4ila5ts), and S10 (https://biorender.com/w0f87at). This material is based upon work supported by the National Science Foundation Graduate Research Fellowship under Grant No. DGE 2146752 (received by S.N.T.). This work was part of the DOE JBEI (https://www.jbei.org) supported by the U.S. Department of Energy, Office of Science, Office of Biological and Environmental Research, supported by the U.S. Department of Energy, Energy Efficiency and Renewable Energy, Bioenergy Technologies Office, through contract DE-AC02-05CH11231 between Lawrence Berkeley National Laboratory and the U.S. Department of Energy (received by S.N.T., M.J.S., A.L., S.S., L.D.K., K.D., S.A., L.C., L.M.W., S.L., L.H., S.K., E.A.T., E.B., A.E., M.G.T., and P.M.S). The funders had no role in manuscript preparation or the decision to publish. The views and opinions of the authors expressed herein do not necessarily state or reflect those of the United States Government or any agency thereof. Neither the United States Government nor any agency thereof, nor any of their employees, makes any warranty, expressed or implied, or assumes any legal liability or responsibility for the accuracy, completeness, or usefulness of any information, apparatus, product, or process disclosed, or represents that its use would not infringe privately owned rights. The United States Government retains, and the publisher, by accepting the article for publication, acknowledges that the United States Government retains a nonexclusive, paid-up, irrevocable, worldwide license to publish or reproduce the published form of this manuscript, or allow others to do so, for United States Government purposes. The Department of Energy will provide public access to these results of federally sponsored research in accordance with the DOE Public Access Plan (http://energy.gov/downloads/doe-public-access-plan).

## Author contributions

P.M.S. and M.G.T. conceptualized the initial study and are the corresponding authors. M.G.T., S.A., L.M.W., and S.N.T. designed

experiments. S.N.T., K.D., and L.C. generated plasmids and strains. S.K., E.A.T., A.E., and E.B. performed the PDC extractions and metabolomics, and S.N.T, S.S., L.D.K., S.L., M.G.T., M.J.S., K.D., A.L., and L.H. performed all other experiments. M.J.S. performed mixed-effect modeling and Monte Carlo simulations, and S.N.T. completed all other data analysis. S.N.T. wrote the draft manuscript. All authors discussed the results, reviewed the article, and approved the final article.

## Competing interests

M.G.T., M.J.S., and P.M.S. have a financial interest in BasidioBio. P.M.S. also has a financial interest in Totality Biosciences. M.G.T., M.J.S., and P.M.S. have no other, non-financial competing interests. All other authors declare that there are no competing interests.
