## [Peer Review File · Nature Communications]

Causes and consequences of experimental variation in *Nicotiana benthamiana* transient expression

Corresponding Author: Dr Patrick Shih

Version 0:

Reviewer comments:

Reviewer #1

(Remarks to the Author)

This manuscript thoroughly and robustly describes variability associated with transient expression of fluorescence proteins in the model plant *Nicotiana benthamiana* using a unique multi-year dataset. This is an important paper for the *N. benthamiana* and plant synthetic biology communities as previous studies have indicated at substantial variability with transient expression but no one has yet systematically explored it as this manuscript does. The authors develop a mixed effect model to attribute variability and test several other possible sources of variability. They go on to test various normalization strategies and conclude by developing a model to estimate the number of plants and leaf discs needed to detect a given effect size.

Minor concerns:

The authors' cite ref 31 Bashandy et al <https://doi.org/10.1186/s13007-015-0091-5> which found most of the variation was due to differences between the sampling spots in *N. benthamiana* leaves. Bashandy et al did not find a consistent pattern in their data but cite Buyel and Fischer <https://doi.org/10.1002/bit.24523> who found certain areas of the leaf gave higher expression. This hypothesis of within-leaf variation should be included in the mixed effects model if possible and follow-up experiment(s) performed to determine if it is a potential contributor to the 41.5% of unexplained variation as done in Fig S2-S5.

The most compelling result in this manuscript is that the best normalization strategy was to use an identical promoter for both fluorescent proteins. The authors point out (line 307-309) that this is "particularly limiting given the popularity of *N. benthamiana* to characterize the effects of promoter identity and architecture on transcription". It seems plausible that some of the experimental-replicate-to-experimental-replicate variability could involve different levels of endogenous *N. benthamiana* transcription factors (TF) depending on variation in growth condition. These TFs drive expression of the *Arabidopsis* PCONs promoters and using the same promoter minimizes the effect of different TF levels between plant batches. This suggests that perhaps some other promoters besides PCM2 might have a more consistent expression level across this variability? Perhaps there is a useful analogy here with choosing qPCR reference genes that have minimal transcriptional changes across many different conditions (e.g. <https://doi.org/10.1038/s41598-018-38247-2>, <https://doi.org/10.1371/journal.pone.0046451>). The broad utility and general interest of this manuscript would be improved if the authors' data could support a recommendation for a normalization promoter to use for comparing different promoters of unknown function or if they test a mechanistic hypothesis explaining why identical promoters work best.

Why did the authors switch the promoter driving eGFP from PCM2 (Figures 2-3) to PCL2 (Figure 5)?

tOcs is described as a "a 722bp 681 spacer" in the caption of Figure 2. If this sequence is the octopine synthase terminator from *Agrobacterium tumefaciens* then some of the differences in part orientation may be explained by "double terminators" formed in some expression cassettes (see Damos and Mason 2018 doi: 10.1111/pbi.12931). This possibility should be mentioned and could optionally be explored to improve the manuscript.

Comments on claims:

The authors claim in the abstract to "develop a comprehensive model of variation in *N. benthamiana* transient expression,

using power analysis to determine the number of individual plants required for a given effect size” but later point out (line 342-346) that “we were nonetheless only able to capture variation in *N. benthamiana* transient expression from a rather narrow perspective. All of our data are from fluorescent reporters from 4-week-old plants, with data collected three days after infiltration. We demonstrated that *Agrobacterium* strain, promoter choice, and T-DNA design all impact variation, but many more strains, plasmids, and expression cassette designs remain untested.”. In light of these understandable limitations, I would suggest removing the word “comprehensive” from this claim or further justify the comprehensive nature of the model, perhaps by extending to alternate reporters beyond fluorescent proteins.

I was confused by line 111 which reads “With these additional experiments, we have accounted for the majority of variation in our model and ruled out other potential sources.” since it suggests that the additional experiments give insight to the remaining 41.5% of variation. The additional experiments do not explain the residual variation because 1) the methods section (lines 420-424) states all the data in the manuscript (except for Fig S2/S3) was collected with 4 week old plants and 350 μ L volume; if this is true then these two factors do not explain residual variation in Figure 1. 2) The methods section does not describe a consistent time of day or time elapsed between collection and measurement) but Figure S4 and S5 indicate no effect of fluorescence values, therefore, these two factors do not explain residual variation in Figure 1. I appreciate and agree with line 108-109 that “Minor changes to water volume (e.g., evaporation, pipetting error) may have contributed to residual error”. One alternate form of line 111 could be “With these additional experiments, we have ruled out other potential sources of variation since collection time has no effect on eGFP variation and water volume and plant age were consistent across the 15 experimental replicates”.

Comments on methods:

The early part of the manuscript is generally clear on which sets of data are incorporated into each CV value presented. For Figure 6, the definition of per-plant CV is clear and is plotted in Fig 6A and 6B. It is unclear how the per-experiment CV contributed the Monte Carlo simulation and the text comment (line 264) on distinction between per-plant and per-experiment CVs was unclear.

While we applaud the authors for posting their code to github, the detail the methods section for the mixed effects model and Monte Carlo simulation was incomplete and additional details would help the reader understand the key parameters in running and developing these models.

The authors should describe in their methods the diameter of the leaf disc, the make/model of the microtiter plate used for measuring leaf disc fluorescence, and the total volume of the each plate well; a small diagram/picture might also help in understanding Figure S2.

Line 409 of the manuscript reads "A highly detailed protocol of *N. benthamiana* growth and care is provided in Supplemental Methods 1" but this file was not available for review. We are delighted that the authors plan to include these details and encourage them to do so.

Supplementary Table 1 is useful but incomplete. This reviewer attempted to use confirm the sequence of several plasmids at <https://public-registry.jbei.org/entry/> but found that the name listed in Supplementary Table 1 was not unique and did not correlate with a specific entry. For example, my guess is that binary vector #1 “PCM2:GFP_CaMV35S2:nptII” is actually JPUB_021048 nicknamed “pCM2-GFP”; I could not find an entry for binary vector #13 BBR1 PCM2:GFP. Please include JPUB part numbers and web links in Supplementary Table 1

It is assumed that the “GFP reporter” described in line 88 and Figure 1 is binary vector #1 = UTR-nptII-CaMV35S_PCM2-eGFP-T_AtUbq3 (nicknamed PCM2:GFP_CaMV35S2:nptII in Supplementary Table 1); this should be stated more explicitly. The UTR adjacent to nptII should be better defined in Figure S1; is this a terminator sequence?

Similarly, the “GFP reporter” described in line 257 and use to create Figure 6 was not defined. (perhaps it is also binary vector #1?). The authors hint at additional data on the 1813 plants (line 259) but there were no additional details in the supplement that I found.

Comments on text and figure clarity:

The term replicate is ambiguous when describing Figure 1 since replicate can refer to 1) experimental replicates (the same experiment run on different days with a different batch of plants), biological replicates (different plants or leaf discs) or technical replicates (repeated measure of the same leaf disc sample). Please use the full phrase “experimental replicate” as often as possible to avoid confusion. I also found “date” to be somewhat unclear in describing differences between experiments run on different days with a different batch of plants. I suggest “experiment date” or “plant batch” replace “date” in the terminology.

To understand this manuscript (especially Figure 4), it is extremely helpful to apprehend the nomenclature of Zhou et al where L=low, M=medium, and H=high. Please make this more obvious to the reader to improve readability of this manuscript. For example, line 132 currently reads “All FP genes are driven by the PCM2 promoter and are terminated by the T_AtUbq3 133 terminator (Supplementary Table 1, Fig. S1)” but could be modified to “All FP genes are driven by the “medium strength” PCM2 constitutive promoter derived from the *Arabidopsis thaliana* HTR5 histone gene (AT4G40040) (see Zhou et al 2023) and are terminated by the *Arabidopsis thaliana* Ubq3 terminator (Supplementary Table 1, Fig. S1)”. [One could underline the letter M in medium and PCM2 to draw attention to the nomenclature]. This more detailed description of

the reporter early in the manuscript is important for interpreting the context of the results. To aid understanding of Figure 4, it could perhaps be useful to underline the L,M,H letters, color the L,M,H letters with red/yellow/green, adding "low", "med", "high" words to the figure, or other similar options.

Readability of Figure 4 and S11 could be improved if individual data boxes of "same promoters" in panels A/B were outlined in bold black line and the meaning indicated by a key rather than labeling A-D with "same promoter" and E-H with "different promoter"

It is assumed that the diamonds on the box plots indicate outlier values beyond 1.5 times the IQR. Please confirm and include this detail in the text and figure captions as appropriate.

Figure 1 caption line 665 would be slightly clearer if "n=64 discs" was replaced with "n=64 discs from 8 separate plants"

Figure 2 instead of "unnormalized", it might be clearer to say "single infiltration"

(Remarks on code availability)

Code has been uploaded to https://github.com/shih-lab/benthi_variation/tree/main. There is no README file that I found. The raw data is uploaded but is not labelled in a manner that helps the reader correlate a given data file with a manuscript figure.

I did not attempt to install and run the code.

Reviewer #2

(Remarks to the Author)

Tang et al. studied variability in the levels of fluorescent proteins transiently expressed in *Nicotiana benthamiana* leaves. They test how different factors affect the fluorescence readout and contribute to variation. Furthermore, they test the efficacy of various normalization schemes to reduce variability and develop a model for statistical power analysis.

The article suffers from two major caveats:

1. The study focuses only on fluorescence readouts and it is not clear how the findings apply to other assays. Similarly, most tested variables affect the readout and/or variability in complex, unpredictable ways. Therefore, the findings from this study cannot be readily applied to experiments that do not follow exactly the protocol used here. The take-home message of the paper then boils down to: "Many different factors affect transgene expression levels and variability in *N. benthamiana*; for more details you need to systematically test your own system".

2. The description of the statistical analysis, especially the Monte Carlo simulation and power analysis are lacking crucial details. The authors state that the dataset comprised "1,813 plants infiltrated with a GFP reporter over nearly three years, spanning experiments conducted by multiple researchers and diverse conditions, including different inoculum densities and *Agrobacterium* strains" (lines 257-259). Did all these additional variables get integrated into the analysis or just the per-plant CV? How meaningful is such a dataset for experiments that will likely be performed with more controlled variables? The authors mention "batches of plants for which at least 30 plants were used" (lines 262-263) but never define what constitutes a "batch". Next, they mention a "Monte Carlo simulation of assay variability" (line 264) but it is unclear what was simulated (per-plant CVs?) and how. For the power analysis, the authors write that they incorporated "additional noise that approximates within-experiment heterogeneity between plants" (lines 273-274) to simulate CVs for individual plants. How much noise was added? Is this noise based on empirical data, and if so, what data? The authors then compare conditions "at varying effect size differences and samples sizes" (lines 276-277). What values or range of values were tested for effect and sample sizes? How many total comparisons were performed? Finally, the authors show results for "GV3101" and "GV3101 under optimal normalization" (line 283) without any explanation as to what this means. Without all these details it is impossible to judge the quality and applicability of the power analysis. This lack of clarity is further compounded by discrepancies between Fig. 6C and 6D. According to Fig. 6D and the text, "The smallest detectable effect size using 50 plants for unnormalized EHA105 was 15.9%" (lines 281-282). However, Fig. 6C indicates that less than 40 plants are sufficient to detect a 15% effect size difference with EHA105. Similarly, to detect a 30% difference with EHA105, you need 10 or 12 plants according to Fig. 6D or 6C, respectively.

In addition to these point, there are a couple of more minor issues:

1. The authors use a mixed-effect model to estimate the effect of replicate-to-replicate, plant-to-plant, and leaf-to-leaf differences on the observed variation. Like the power analysis, this needs more details. How was this model implemented? What parameters were used? Were replicate, plant, and leaf the only variables used by the model?

2. Related to the above point, at the end of the mixed-effect model analysis, the authors conclude that they "have accounted for the majority of variation in [their] model and ruled out other potential sources". How, then, do they explain the missing ~40% of variability? There are countless additional potential sources of variability (e.g. the researcher conducting the experiment, the preparation of the *Agrobacteria* used for transformation, the plate reader used, the position on the plate, the order in which the samples were collected, ...). It is impossible to rule them all out.

3. On lines 154-155, the authors write that "Few schemes produce an eGFP/mCherry CV that is significantly less than

scheme 1's eGFP CV following a Bonferroni correction". However, according to Fig. 2B, most (10 out of 17) schemes match this description.

4. In the discussion on lines 317-318, the authors mention the lack of a published analysis of how the design of multi-gene cassettes affects transgene expression. This was at least to some extent done in Kallam et al., 2023, Plant Biotech. J., doi: 10.1111/pbi.14048

5. In the methods section, more details are needed for the fluorescence measurement of the leaf discs. Which side of the leaf discs was facing up in the 96-well plate? What were the gain settings? Were transparent or opaque plates used?

(Remarks on code availability)

Version 1:

Reviewer comments:

Reviewer #1

(Remarks to the Author)

This manuscript is substantially improved and the authors have addressed many of my comments appropriately.

Minor concern:

The authors' added analysis in Supplementary Figure S2 is helpful and the new data on RUBY and PDC in Supplementary Figure S8 is welcome. However, it is unclear how multiplicative scatter is calculated. I made effort to understand this but could find almost no references or tutorials (nearly all of them instead refer to multiplicative scatter correction). The author's in-caption description in Figure S2 is helpful, but insufficient: "Multiplicative scatter indicates the factor by which the spread of a distribution around the mean is increased by a particular component of the variance, if all other factors are held constant. Multiplicative scatter of 1 indicates that the component contributes no additional variance." Please cite an appropriate source that readers can use to understand this statistical approach or (recommended) include a brief explanation of how this value is calculated.

(Remarks on code availability)

Reviewer #2

(Remarks to the Author)

The authors have addressed my concerns and I have no further comments.

(Remarks on code availability)

REVIEWER COMMENTS

Reviewer #1 (Remarks to the Author):

This manuscript thoroughly and robustly describes variability associated with transient expression of fluorescence proteins in the model plant *Nicotiana benthamiana* using a unique multi-year dataset. This is an important paper for the *N. benthamiana* and plant synthetic biology communities as previous studies have indicated substantial variability with transient expression but no one has yet systematically explored it as this manuscript does. The authors develop a mixed effect model to attribute variability and test several other possible sources of variability. They go on to test various normalization strategies and conclude by developing a model to estimate the number of plants and leaf discs needed to detect a given effect size.

Minor concerns:

The authors' cite ref 31 Bashandy et al <https://doi.org/10.1186/s13007-015-0091-5> which found most of the variation was due to differences between the sampling spots in *N. benthamiana* leaves. Bashandy et al did not find a consistent pattern in their data but cite Buyel and Fischer <https://doi.org/10.1002/bit.24523> who found certain areas of the leaf gave higher expression. This hypothesis of within-leaf variation should be included in the mixed effects model if possible and follow-up experiment(s) performed to determine if it is a potential contributor to the 41.5% of unexplained variation as done in Fig S2-S5.

We thank the reviewer for bringing Buyel and Fischer to our attention, and we now reference this publication in the introduction.

We have fully revised our model to better account for variation within this assay. Previously, the difference between leaves was modeled as a fixed effect due to a general trend of younger leaves having higher transient expression than older ones. While this trend holds true on the population level, analysis of individual plants shows numerous cases where leaf-level data are comparable between leaves, or on occasion, where older leaves have higher expression than their younger counterparts. Incorporating variance between leaves as a dynamic mixed effect enabled better classification of residual variance, with 28.2% of variation being attributable to leaf-to-leaf variance in our data set.

Additionally, we conducted an experiment to evaluate variation that occurs from discs sampled from the same portion of a leaf, discs from different portions of a leaf (proximal to the petiole and distal), and technical variance from the plate reader. This allowed the residual variance to be better parsed and allowed for 22.3% of the variation to be attributed to different discs collected in the same leaf location on a plant.

When comparing different leaf positions, we observed significant positional effects on leaf T5 but not T4 for GV3101. When using an additional strain, EHA105, we found significant effects for both leaves, demonstrating leaf and strain effects for leaf-level positional variance. These new results

have been incorporated into Supplemental Fig. S5. As all data shown in Figure 1 were collected at a distal site on the leaf, this was a controlled variable for our multi-year dataset, but these additional results highlight a potentially important source of variability for some experiments.

The most compelling result in this manuscript is that the best normalization strategy was to use an identical promoter for both fluorescent proteins. The authors point out (line 307-309) that this is “particularly limiting given the popularity of *N. benthamiana* to characterize the effects of promoter identity and architecture on transcription”. It seems plausible that some of the experimental-replicate-to-experimental-replicate variability could involve different levels of endogenous *N. benthamiana* transcription factors (TF) depending on variation in growth condition. These TFs drive expression of the *Arabidopsis* PCONS promoters and using the same promoter minimizes the effect of different TF levels between plant batches. This suggests that perhaps some other promoters besides PCM2 might have a more consistent expression level across this variability? Perhaps there is a useful analogy here with choosing qPCR reference genes that have minimal transcriptional changes across many different conditions (e.g. <https://doi.org/10.1038/s41598-018-38247-2>, <https://doi.org/10.1371/journal.pone.0046451>). The broad utility and general interest of this manuscript would be improved if the authors’ data could support a recommendation for a normalization promoter to use for comparing different promoters of unknown function or if they test a mechanistic hypothesis explaining why identical promoters work best.

The pipeline to identify the promoters that make up the PCONS library takes into account (1) their ubiquity of expression across different tissue types and (2) their low variation in expression. This was possible due to the availability of an RNA-seq *Arabidopsis* tissue atlas (Klepikova et al), but there is no such equivalent for *N. benthamiana*. We used these promoters precisely because we also believed at the outset that we should select promoters with minimal transcriptional changes across many different conditions, but our data indicate that there is noise inherent to the system that cannot be totally eliminated, even after curating lists of genes with relatively invariant expression. We contend that one of the most important takeaways is that there *cannot* be a universal normalizing promoter, as that would require a promoter that co-varies with all other promoters under all conditions, which cannot be true.

Since there is a dearth of knowledge about which transcription factor(s) regulate any promoter in plants and how, it is very difficult to test mechanistically why different promoter combinations do or do not co-vary. It also cannot be done exhaustively since the complete set of TFs that act upon any promoter in any plant is unknown. As a community, we are only just beginning to gain such genome-scale and functional genomic data in *Arabidopsis*, with dramatically less known in *N. benthamiana*. We agree that this is a fascinating question that we intend to pursue over the next several years when it is possible to better dissect the question of normalization, but currently we lack the knowledge and resources to carry out a comprehensive gene co-expression analysis and believe it is better suited in a follow up study.

Why did the authors switch the promoter driving eGFP from PCM2 (Figures 2-3) to PCL2 (Figure 5)?

Since PCL2 yielded the lowest average CV (between eGFP/mCherry CV and mCherry/eGFP CV) when driving both FPs (Fig. 4, Fig. S11), we proceeded using that promoter. This decision is now explicitly justified in-text: “To explore whether normalization makes comparisons between experimental replicates or to historical data more valid, we co-infiltrated GFP driven by PCL2 (which yielded the lowest average normalized CVs when driving both FPs (Fig. 4, S11)) alone and with all six mCherry binary vectors thus far generated in six independent experimental replicates.”

tOcs is described as a “a 722bp 681 spacer” in the caption of Figure 2. If this sequence is the octopine synthase terminator from *Agrobacterium tumefaciens* then some of the differences in part orientation may be explained by “double terminators” formed in some expression cassettes (see Diamos and Mason 2018 doi: 10.1111/pbi.12931). This possibility should be mentioned and could optionally be explored to improve the manuscript.

The formation of a double terminator very well may increase the expression of the doubly terminated gene, and we have now mentioned the possibility in the main text: “The formation of a double terminator with the tOcs spacer in some cassettes may be partly responsible for increased expression, as has been shown with other binary vectors, but reversing the direction of tOcs does not consistently increase fluorescence from a doubly terminated cassette over the corresponding singly terminated cassette (Fig. S10).”

To explore the effect of the double terminator, we generated three new binary vectors identical to the binary vectors in normalization schemes 8, 9, and 10 except with tOcs reversed (henceforth referred to as 8R, 9R, and 10R). A double terminator is lost or shifted to the other cassette when comparing these normalization schemes to the “reversed” counterpart. We infiltrated 36 plants and measured their eGFP and mCherry fluorescence signals (6 plants per binary vector, 2 leaves per plant, 4 discs per leaf). Supplementary Figure S7 has been updated to include these new findings. One-tailed Student’s t-tests to determine whether the doubly terminated cassette is more highly expressed than the singly terminated cassette revealed that this is the case in only half of the tested instances. While double termination may contribute to the patterns we have observed, it is not sufficient to explain everything.

Comments on claims:

The authors claim in the abstract to “develop a comprehensive model of variation in *N. benthamiana* transient expression, using power analysis to determine the number of individual plants required for a given effect size” but later point out (line 342-346) that “we were nonetheless only able to capture variation in *N. benthamiana* transient expression from a rather narrow perspective. All of our data are from fluorescent reporters from 4-week-old plants, with data collected three days after infiltration. We demonstrated that *Agrobacterium* strain, promoter choice, and T-DNA design all impact variation, but many more strains, plasmids, and expression cassette designs remain untested.”. In light of these understandable limitations, I would suggest

removing the word “comprehensive” from this claim or further justify the comprehensive nature of the model, perhaps by extending to alternate reporters beyond fluorescent proteins.

While our updated model accounts for practically all variance, there are assumptions of controlled plant age, leaf identity (T4 and T5), leaf position, and use of the GV3101 strain. We have therefore removed “comprehensive” from line 33 to better reflect the limitations of this study.

I was confused by line 111 which reads “With these additional experiments, we have accounted for the majority of variation in our model and ruled out other potential sources.” since it suggests that the additional experiments give insight to the remaining 41.5% of variation. The additional experiments do not explain the residual variation because 1) the methods section (lines 420-424) states all the data in the manuscript (except for Fig S2/S3) was collected with 4 week old plants and 350 μ L volume; if this is true then these two factors do not explain residual variation in Figure 1. 2) The methods section does not describe a consistent time of day or time elapsed between collection and measurement) but Figure S4 and S5 indicate no effect of fluorescence values, therefore, these two factors do not explain residual variation in Figure 1. I appreciate and agree with line 108-109 that “Minor changes to water volume (e.g., evaporation, pipetting error) may have contributed to residual error”. One alternate form of line 111 could be “With these additional experiments, we have ruled out other potential sources of variation since collection time has no effect on eGFP variation and water volume and plant age were consistent across the 15 experimental replicates”.

As mentioned in the first comment, we have expanded the model to account for almost all residual variance. By breaking down variance into the following categories: week-to-week mean shifts, week-to-week intraplant variation, plant-to-plant mean variation, leaf-to-leaf variation, disc-to-disc variation, and technical/residual variation, we were able to account for >99% of variation within our dataset. There are previously mentioned assumptions here regarding plant age, leaf identity, leaf position, and strain (which were all held constant in our dataset), but our new model performs substantially better than the previous form. These updates have been incorporated into Fig. 1D.

Comments on methods:

The early part of the manuscript is generally clear on which sets of data are incorporated into each CV value presented. For Figure 6, the definition of per-plant CV is clear and is plotted in Fig 6A and 6B. It is unclear how the per-experiment CV contributed the Monte Carlo simulation and the text comment (line 264) on distinction between per-plant and per-experiment CVs was unclear.

We have addressed the construction of the Monte Carlo simulation in detail below, explaining how per-plant and per-experiment CVs were incorporated into the model.

While we applaud the authors for posting their code to github, the detail the methods section for the mixed effects model and Monte Carlo simulation was incomplete and

additional details would help the reader understand the key parameters in running and developing these models.

We have now provided information on how the Monte Carlo simulation and Mixed Effects Model were constructed in Supplementary Methods 2. The new description of the simulation and modeling in this file is reproduced below:

Monte Carlo Simulation

A dataset of 1,813 *N. benthamiana* plants from 32 independent GFP transient expression experiments was compiled, spanning multiple years, researchers, and binary vector designs using either *A. tumefaciens* GV3101 (1,087 plants) or EHA105 (726 plants). This dataset captures the real-world variability of transient expression performance outside of a single controlled experimental environment.

For each plant, a coefficient of variation (CV) of GFP expression was calculated from eight leaf disks. Using these data, we parameterized a hierarchical Monte Carlo model to reproduce observed variability at three levels: (i) week-to-week shifts in average plant CV, representing overall plant quality between experiments; (ii) plant-to-plant heterogeneity within a given week; and (iii) within-plant disk-level noise.

Weekly median plant CVs followed a log-normal distribution with parameters $\text{meanlog} = -1.447$, $\text{sdlog} = 0.300$ for GV3101 and $\text{meanlog} = -0.903$, $\text{sdlog} = 0.460$ for EHA105. The within-week spread of plant CVs (weekly SD-of-CV) also followed a log-normal distribution ($\text{meanlog} = -2.3248$, $\text{sdlog} = 0.3897$). Fit adequacy was confirmed using Kolmogorov–Smirnov and density RMSE metrics.

These data were used to simulate variance across random weeks. For each simulated experiment, a week quality value was selected using a random number from the log-normal distribution of the strain of choice (GV3101 vs EHA105), representing the average plant CV for that week. Additionally, a week quality spread value—modeled from the empirical standard deviation of plant CVs across different experiments—was randomly selected from the underlying log-normal distribution and used for the standard deviation of CV values for that week. These two values were then used to generate individual plant CVs for the simulated experiment, incorporating both the mean and standard deviation values for plant CV that were randomly generated for that week. Each plant drawn from this distribution thus has its own unique CV value, bounded by biologically relevant values between 0.03-2.0, which were inferred from the 1,813 measured plant spread.

For each simulated plant, 8 leaf discs were generated using an arbitrary mean value of 100,000 GFP units and standard deviation derived from the plant CV. A given simulated experiment would then have all leaf disc data bulked (e.g. 10 plants with 80 total discs) and used for population-level comparisons. For the modeling in this study to determine features such as minimal detectable effect size or power analyses, 1000 weeks of experiments were simulated using the

above pipeline. Simulation outputs were summarized as statistical power (fraction of iterations with significant construct differences) across plant counts and effect sizes. Comparison of features such as effect size was conducted through two-sided t-tests with Benjamini–Hochberg FDR control at $\alpha=0.05$. For power analysis modeling, a $\geq 95\%$ accurate detection of differences was used as a threshold for classifying sufficient detection power.

To validate the outputs of this simulation, goodness-of-fit between simulated and empirical CV distributions was assessed using KS and 1D Wasserstein distances (Fig. 6b).

Mixed effect modeling

Modeling of variance was conducted using the pCM2::GFP dataset which consisted of GFP expression data from 114 plants collected over 15 weeks. Fluorescence data ($n = 8$ discs per plant) were transformed to a \log_{10} scale to linearize multiplicative effects and to account for heteroscedasticity.

On this scale, a linear mixed-effects model was fit to account for 4 major sources of variation: (i) experiment-to-experiment shifts in plant mean, (ii) experiment-to-experiment shifts in intraplant variance, (iii) plant-level shifts in mean GFP expression, and (iv) leaf-to-leaf expression variability, along with a residual component. The equation for modeling such variance for the GFP expression of a given leaf disc is as follows:

$$\begin{aligned} \text{Log}_{10}\text{GFP}(\text{disc, plant, week}) = & \\ & \beta_0 \text{ (dataset mean) +} \\ & \beta_1 * \text{Leaf position (dataset difference between top and bottom leaf) +} \\ & u_0 \text{ (week's deviation in mean GFP) +} \\ & u_1 * \text{Leaf position (weeks deviation in top vs bottom leaf difference) +} \\ & v_0 \text{ (Individual plant's mean deviation) +} \\ & v_1 \text{ (Individual plant's leaf-to-leaf variation) +} \\ & e \text{ (residual error)} \end{aligned}$$

This was calculated using the R code:

```
m_final <- lmer(
  log_GFP ~ Leaf_c +
  (1 + Leaf_c | Date) +
  (1 + Leaf_c | Unique_plant_ID),
  data = PC4_data, REML = TRUE)
```

In the above, Leaf_c encoded the contrast between the top and bottom leaves.

Disc-to-disc, positional, and technical variance:

To determine the impact of disc-to-disc variability, leaf position, and technical replication noise on total observed variability, the same pCM2::GFP construct was used to infiltrate 24 plants on the

same two leaves (T4 and T5) in two positions (proximal to the petiole and distal) with two *Agrobacterium* strains (GV3101 and EHA105, 12 plants per strain). For each leaf, eight discs were taken (four per proximal or distal position). GFP fluorescence was quantified as before, except plates were scanned three times each in their standard position and then were flipped 180 degrees and scanned three more times to enable quantification of technical variance. The following R code when then used to fit a mixed effects model to better account for residual variation:

```
m_leafc <- lmer(
  log_GFP ~
  Plate_Position + (location of disc on the plate) +
  Leaf_c + (Leaf T4 or T5 variance)
  (1 + Leaf_c | Plant) + (Plant mean and leaf-to-leaf variance)
  (1 | Plant:Leaf:Site) + (Proximal vs distal position impact)
  (1 | DiskID) + (disc-to-disc variation),
  data = punchvar, REML = TRUE)
```

Ruby and PDC modeling

Modeling of metabolite production variance was conducted in a similar way but with reduced variables given tissue bulking that was done prior to extraction. For Ruby, a single infiltrated spot per leaf was excised and extracted, resulting in data for only week-to-week and plant-to-plant variability. As only a single measurement was taken per leaf, the residual of this model includes leaf-to-leaf variability, as calculated by the following R code:

```
m_ruby <- lmer(
  log10_ruby ~ 1+
  (1 | Experimental_replicate) + (experiment mean variation)
  (1 | Experimental_replicate:Plant), (plant level variation within experiment)
  REML = TRUE)
```

For PDC production, tissue from both leaves was bulked prior to extraction. For this reason, a linear model was used to examine the relationship between week and PDC production, with residual variance representing plant-to-plant and leaf-to-leaf variability not explained by week-to-week differences.

The authors should describe in their methods the diameter of the leaf disc, the make/model of the microtiter plate used for measuring leaf disc fluorescence, and the total volume of the each plate well; a small diagram/picture might also help in understanding Figure S2.

Additional details about the 96-well plate were added to the Methods section: “The leaf discs were placed abaxial side up on 350 μ L of water in black, 360- μ L 96 well Costar Assay Plates with clear flat bottoms (Corning). eGFP (Ex. λ = 488 nm, Em. λ = 520 nm) and mCherry (Ex. λ =

587 nm, Em.λ = 615 nm) fluorescences were measured using a Synergy 4 microplate reader (Bio-tek). Gain was set at 100 and read height at 10.5 mm.”

Line 409 of the manuscript reads "A highly detailed protocol of *N. benthamiana* growth and care is provided in Supplemental Methods 1“ but this file was not available for review. We are delighted that the authors plan to include these details and encourage them to do so.

We thank the reviewer for notifying us of this oversight. The document Supplemental Methods 1 has now been provided which contains a detailed plant care protocol.

Supplementary Table 1 is useful but incomplete. This reviewer attempted to use confirm the sequence of several plasmids at <https://public-registry.jbei.org/entry/> but found that the name listed in Supplementary Table 1 was not unique and did not correlate with a specific entry. For example, my guess is that binary vector #1 “PCM2:GFP_CaMV35S2:nptII” is actually JPub_021048 nicknamed “pCM2-GFP”; I could not find an entry for binary vector #13 BBR1 PCM2:GFP. Please include JPub part numbers and web links in Supplementary Table 1

All plasmid sequences and strains are now viewable (after making a free account) at this public registry folder: <https://public-registry.jbei.org/folders/929>. The table has also been updated to include part numbers for easier searching in the ICE registry.

It is assumed that the “GFP reporter” described in line 88 and Figure 1 is binary vector #1 = UTR-nptII-CaMV35S_PCM2-eGFP-T_AtUbq3 (nicknamed PCM2:GFP_CaMV35S2:nptII in Supplementary Table 1); this should be stated more explicitly. The UTR adjacent to nptII should be better defined in Figure S1; is this a terminator sequence?

The GFP reporter is now explicitly referred to as vector 1 from Supplementary Table 1 in the text. The “UTR” in Supplementary Fig. S1 is the CaMV 3’ UTR, and the caption of the table now states that explicitly.

Similarly, the “GFP reporter” described in line 257 and use to create Figure 6 was not defined. (perhaps it is also binary vector #1?). The authors hint at additional data on the 1813 plants (line 259) but there were no additional details in the supplement that I found.

A description of the underlying construct has been included in the new methods section outlining the Monte Carlo simulation:

“A dataset of 1,813 *N. benthamiana* plants from 32 independent GFP transient expression experiments was compiled, spanning multiple years, researchers, and binary vector designs using either *A. tumefaciens* GV3101 (1,087 plants) or EHA105 (726 plants).”

As noted above, the GFP expression data from the 1,813 plants were collected from a wide array of experiments, spanning different researchers and binary vector designs. Thus, there is no universal vector used, although all plants did express GFP. We view this as a feature rather than

a bug of the dataset as it enables a more robust modeling of real-world variance that one could expect to see using this assay. Regardless of promoter strength or construct design, all GFP data were standardized to their underlying CV value (standard deviation of expression over 8 discs / mean of expression), resulting in a unitless CV value that can be used for comparison across experiments. The CV values and experimental identifiers from the 1,813 plants have been included as a .csv.

Comments on text and figure clarity:

The term replicate is ambiguous when describing Figure 1 since replicate can refer to 1) experimental replicates (the same experiment run on different days with a different batch of plants), biological replicates (different plants or leaf discs) or technical replicates (repeated measure of the same leaf disc sample). Please use the full phrase “experimental replicate” as often as possible to avoid confusion. I also found “date” to be somewhat unclear in describing differences between experiments run on different days with a different batch of plants. I suggest “experiment date” or “plant batch” replace “date” in the terminology.

All instances of “replicate” meaning “experimental replicate,” when not obvious from the context of the sentence, have been changed to “experimental replicate.” Additionally, the sentence “Dates of measurement are unique identifiers for experimental replicates.” has been removed to avoid confusion between date/batch/experimental replicate.

To understand this manuscript (especially Figure 4), it is extremely helpful to apprehend the nomenclature of Zhou et al where L=low, M=medium, and H=high. Please make this more obvious to the reader to improve readability of this manuscript. For example, line 132 currently reads “All FP genes are driven by the PCM2 promoter and are terminated by the T_AtUbq3 133 terminator (Supplementary Table 1, Fig. S1)” but could be modified to “All FP genes are driven by the “medium strength” PCM2 constitutive promoter derived from the Arabidopsis thaliana HTR5 histone gene (AT4G40040) (see Zhou et al 2023) and are terminated by the Arabidopsis thaliana Ubq3 terminator (Supplementary Table 1, Fig. S1)”. [One could underline the letter M in medium and PCM2 to draw attention to the nomenclature]. This more detailed description of the reporter early in the manuscript is important for interpreting the context of the results. To aid understanding of Figure 4, it could perhaps be useful to underline the L,M,H letters, color the L,M,H letters with red/yellow/green, adding “low”, “med”, “high” words to the figure, or other similar options.

The aforementioned line has been edited according to the reviewer’s suggestion: “All FP genes are driven by the medium strength PCM2 constitutive promoter derived from the *A. thaliana* HTR5 histone gene (AT4G40040) and are terminated by the *A. thaliana* Ubq3 terminator (Supplementary Table 1, Fig. S1).”

Explicit mention of the low/medium/high strength constitutive promoter nomenclature (PCL/PCM/PCH) is also now made in the text: “From a library of low, medium, and high strength constitutive promoters characterized by Zhou et al. (annotated as “PCL”, “PCM”, and “PCH,”

respectively), we selected low- and high-strength promoters, PCL2 and PCH5, to test alongside PCM2.”

Readability of Figure 4 and S11 could be improved if individual data boxes of “same promoters” in panels A/B were outlined in bold black line and the meaning indicated by a key rather than labeling A-D with “same promoter” and E-H with “different promoter”

The intent behind the terms “same promoters” or “different promoters” was to indicate that the 3 promoters on the mCherry axis and the eGFP axis were either the same set of 3 promoters or a different set of 3 promoters. Figure 4 labels have been modified to “Same Promoter Set” and “Different Promoter Set”, and additional emphasis on these terms were added to the text to clarify the difference between the sets and “identical promoters,” which refers to cassettes with identical promoter sequences:

“eGFP and mCherry were driven by this same set of three promoters (“same promoter set”).”
“We then sought to clarify whether the large decrease in CV when using identical promoters is due to the similar promoter strengths or to other variables, such as shared trans factors affecting mRNA abundance. To do so, we generated another set of three binary vectors with mCherry driven by PCL1, PCM1, and PCH4 (“different promoter set”), which are the promoters in the library closest in expression to the three already tested.”

It is assumed that the diamonds on the box plots indicate outlier values beyond 1.5 times the IQR. Please confirm and include this detail in the text and figure captions as appropriate.

All boxplots with outliers beyond 1.5*IQR are now captioned appropriately.

Figure 1 caption line 665 would be slightly clearer if “n=64 discs” was replaced with “n=64 discs from 8 separate plants”

Figure 1 caption has been edited according to the reviewer’s suggestion.

Figure 2 instead of “unnormalized”, it might be clearer to say “single infiltration”

To avoid confusion with BiBi infiltrations (which are also single infiltration in the sense that a single strain is infiltrated), we prefer the term unnormalized.

Reviewer #1 (Remarks on code availability):

Code has been uploaded to https://github.com/shih-lab/benthi_variation/tree/main. There is no README file that I found. The raw data is uploaded but is not labelled in a manner that helps the reader correlate a given data file with a manuscript figure.

I did not attempt to install and run the code.

All data are annotated with the relevant metadata in the jupyter notebook `tangetal2025_code.ipynb`, and we have commented the jupyter notebook extensively to make clear how the figures are generated and with which data. For additional clarity, a README file has been added to the submission materials folder and to the GitHub.

Reviewer #2 (Remarks to the Author):

Tang et al. studied variability in the levels of fluorescent proteins transiently expressed in *Nicotiana benthamiana* leaves. They test how different factors affect the fluorescence readout and contribute to variation. Furthermore, they test the efficacy of various normalization schemes to reduce variability and develop a model for statistical power analysis.

The article suffers from two major caveats:

1. The study focuses only on fluorescence readouts and it is not clear how the findings apply to other assays. Similarly, most tested variables affect the readout and/or variability in complex, unpredictable ways. Therefore, the findings from this study cannot be readily applied to experiments that do not follow exactly the protocol used here. The take-home message of the paper then boils down to: "Many different factors affect transgene expression levels and variability in *N. benthamiana*; for more details you need to systematically test your own system".

In order to address the concern that the trends observed in fluorescence from FPs might not translate to other, potentially more complex assays/readouts, we looked into variation in yield for two different metabolic pathways: betalain (delivered as three enzymes on separate, co-infiltrated T-DNAs or as three enzymes connected by T2A self-cleaving peptides in a single T-DNA, i.e., the RUBY reporter) and 2-pyrone-4,6-dicarboxylic acid (PDC) (delivered as five enzymes on separate, co-infiltrated T-DNAs). We performed three experimental replicates of the betalain pathway and four experimental replicates of the PDC pathway (n=6 plants) on separate dates with different plant batches, and the findings are summarized in the new Supplementary Fig. S8.

A Bonferroni-corrected Student's t-test revealed that one of the three betalain replicates was significantly different from the other two, for both the co-infiltration and T2A. For co-infiltrated samples, the absorbances of the highest and lowest yielding replicates were only about ~2.3-fold different, but for the T2A samples, the ~3.9-fold difference was as much as the differences we observed for FPs. Yields of the PDC replicates were not significantly different after Bonferroni correction, but we observed ~1.2-fold difference in mg PDC per g dry weight between the highest and lowest yielding replicates.

The differences in variation may in part owe to the different methods of extraction for the two metabolites. Discrepancies with the FPs may also be due to the fact that FP fluorescence is a more direct readout of transgene expression than a metabolic pathway, which is subject to rate-limiting steps, enzyme localization, substrate diffusion, etc. A many-step biosynthetic pathway must also contend with heterogeneity in the population of transformed plant cells, which can

receive all T-DNAs or any subset of the infiltrated T-DNAs. If any enzymes or substrates are sequestered in specific cellular compartments rather than freely diffusing within or between (via plasmodesmata) cells, this further complicates the relationship between transgene expression and the ultimate yield of the metabolic pathway. Although variability in total pathway yields may be specific to the particular pathway, we believe that the variability characterized for FP expression translates to the variability in expression of each component transgene in a metabolic pathway.

2. The description of the statistical analysis, especially the Monte Carlo simulation and power analysis are lacking crucial details. The authors state that the dataset comprised "1,813 plants infiltrated with a GFP reporter over nearly three years, spanning experiments conducted by multiple researchers and diverse conditions, including different inoculum densities and Agrobacterium strains" (lines 257-259). Did all these additional variables get integrated into the analysis or just the per-plant CV? How meaningful is such a dataset for experiments that will likely be performed with more controlled variables? The authors mention "batches of plants for which at least 30 plants were used" (lines 262-263) but never define what constitutes a "batch". Next, they mention a "Monte Carlo simulation of assay variability" (line 264) but it is unclear what was simulated (per-plant CVs?) and how. For the power analysis, the authors write that they incorporated "additional noise that approximates within-experiment heterogeneity between plants" (lines 273-274) to simulate CVs for individual plants. How much noise was added? Is this noise based on empirical data, and if so, what data? The authors then compare conditions "at varying effect size differences and samples sizes" (lines 276-277). What values or range of values were tested for effect and sample sizes? How many total comparisons were performed? Finally, the authors show results for "GV3101" and "GV3101 under optimal normalization" (line 283) without any explanation as to what this means. Without all these details it is impossible to judge the quality and applicability of the power analysis. This lack of clarity is further compounded by discrepancies between Fig. 6C and 6D. According to Fig. 6D and the text, "The smallest detectable effect size using 50 plants for unnormalized EHA105 was 15.9%" (lines 281-282). However, Fig. 6C indicates that less than 40 plants are sufficient to detect a 15% effect size difference with EHA105. Similarly, to detect a 30% difference with EHA105, you need 10 or 12 plants according to Fig. 6D or 6C, respectively.

Monte Carlo and Power Analysis Details

A detailed description of the Monte Carlo simulation and Power Analysis has been included in the Supplementary Methods 2 file, as well as above in response to Reviewer 1. This new section should answer many of the questions in this comment, particularly those related to how diverse GFP expression data were used to estimate variance that occurs on the per-week and per-plant levels.

“How meaningful is such a dataset for experiments that will likely be performed with more controlled variables?”

The dataset used to develop the Monte Carlo simulation contained GFP expression data from 1,813 plants and was derived from 32 separate experiments. These experiments were conducted by multiple researchers under a variety of conditions. We view this as a feature of our data rather

than a bug. When we began this study, we hoped to create a model of transient expression variance that was robust and broadly applicable. By incorporating these diverse experiments, we attempted to capture real-world variance that would likely be encountered beyond our lab and beyond a single controlled construct.

We acknowledge that the variance observed in a tightly controlled experimental setting may be lower than the variance predicted by our model under certain conditions. However, when estimating sample size for sufficient statistical power without *a priori* knowledge of construct performance, it is often advisable to adopt a conservative framework that incorporates the range of variability historically observed across experiments. This approach reduces the risk of underpowered studies—an especially important consideration for *N. benthamiana* which is known for high biological and technical variability. In this context, we argue that it is preferable to model on the side of conservatism in power analysis rather than optimism.

With this in mind, we have added the following line to inform readers that the diverse dataset used to train the simulation accounts for historical variation, and therefore, individual controlled results may achieve lower than predicted variances: Because this dataset accounts for historical variation, the simulations trained with it will give conservative estimates. Individual, well-controlled experiments may yield lower variations, but since the precise variation cannot be known beforehand, erring on the side of excessive rather than insufficient statistical power is preferable.

Fig. 6C vs 6D

We thank the reviewer for highlighting this disparity to allow for proper clarification in the figure. For Fig. 6c, a Monte Carlo simulation was used to simulate the output of two constructs with a defined expression difference across 1000 weeks of experiments. For example, for a 20% effect size Construct A would have a mean output of 100,000 units and Construct B 120,000 units, with a standard deviation of output dictated by individual plant CVs. A set number of plants—starting at 2 and counting upward—would be used across the 1000 weeks, and within each week, a comparison of GFP outputs would be made between the two constructs using a t-test. Significant and accurate differentiation between the constructs would be tabulated as an accurate comparison for that week. If at least 95% of simulated weeks (≥ 950 weeks) had an accurate and significant comparison, the underlying plant number was deemed sufficiently powerful to detect the underlying effect size. The smallest possible plant number to achieve the 95% threshold for each stated effect size is shown in Fig. 6C.

For Fig. 6D, a best fitting line was fit to all of the effect-size and plant-number points derived from 6c. As no function was found that perfectly fit these modeled points, there is an associated error with each of these lines, particularly at the extremes of effect size. To demonstrate this, we have included the individual points that we fit the lines to (exponential decay regression). The disparity between figure panels comes from the difference between an exact modeled value and best-fitting equation-derived value. To clarify that Fig. 6C is a direct modeled calculation while Fig. 6D is a best-fitting equation fit to these points, we have added the following sentence within the text: “From these calculations of plants needed for given effect sizes, we fit an exponential decay curve

assuming fixed CV for three situations: unnormalized EHA105, unnormalized GV3101, and optimally normalized GV3101 (lowest CV achieved in this publication) (Fig. 6D).”

In addition to these point, there are a couple of more minor issues:

1. The authors use a mixed-effect model to estimate the effect of replicate-to-replicate, plant-to-plant, and leaf-to-leaf differences on the observed variation. Like the power analysis, this needs more details. How was this model implemented? What parameters were used? Were replicate, plant, and leaf the only variables used by the model?

We have created an updated model to better incorporate leaf-to-leaf variability within a plant. Previously, this variable was held as a fixed effect due to a population-level trend of the younger leaf T4 having higher expression than the older leaf T5. Further investigation found numerous exceptions where leaves T4 and T5 within a plant had equivalent expression, or in some instances, where T5 expression exceeded that of T4. Incorporating this feature as a mixed effect allowed for variance to be classified into four main groups: week-to-week mean shifts, week-to-week plant variance between leaves, individual plant mean variance, and individual plant leaf-to-leaf differences. Additionally, we conducted another experiment to investigate the impact of disc-to-disc variation, leaf position, and technical replication errors. Taking into account the binary vector, *Agrobacterium* strain, plant age, growth conditions, and leaves that were held constant for our modeling dataset, this updated model now accounts for >99% of the observed variance.

A detailed description of how the model was constructed along with the underlying equation and code have now been provided in Supplementary Methods 2.

2. Related to the above point, at the end of the mixed-effect model analysis, the authors conclude that they "have accounted for the majority of variation in [their] model and ruled out other potential sources". How, then, do they explain the missing ~40% of variability? There are countless additional potential sources of variability (e.g. the researcher conducting the experiment, the preparation of the *Agrobacteria* used for transformation, the plate reader used, the position on the plate, the order in which the samples were collected, ...). It is impossible to rule them all out.

With our updated model and additional experiment to investigate disc-to-disc vs technical variability, we have now accounted for >99% of the observed variability as detailed in the new modeling section of the methods.

3. On lines 154-155, the authors write that "Few schemes produce an eGFP/mCherry CV that is significantly less than scheme 1's eGFP CV following a Bonferroni correction". However, according to Fig. 2B, most (10 out of 17) schemes match this description.

We appreciate your bringing this error to our attention. The sentence has been edited to now read: "Most schemes produce an eGFP/mCherry CV that is significantly less than scheme 1's

eGFP CV following a Bonferroni correction (Fig. 2B), but only scheme 3 (co-infiltration of two pVS1 binary vectors) meets this condition when mCherry is treated as the reporter (Fig. 2C)."

4. In the discussion on lines 317-318, the authors mention the lack of a published analysis of how the design of multi-gene cassettes affects transgene expression. This was at least to some extent done in Kallam et al., 2023, Plant Biotech. J., doi: 10.1111/pbi.14048

We thank the reviewer for informing us of this publication, and it is now mentioned in the discussion.

5. In the methods section, more details are needed for the fluorescence measurement of the leaf discs. Which side of the leaf discs was facing up in the 96-well plate? What were the gain settings? Were transparent or opaque plates used?

Methods were edited to now include this text: "The leaf discs were placed abaxial side up on 350 μ L of water in black, 360- μ L 96 well Costar Assay Plates with clear flat bottoms (Corning). eGFP (Ex. λ = 488 nm, Em. λ = 520 nm) and mCherry (Ex. λ = 587 nm, Em. λ = 615 nm) fluorescences were measured using a Synergy 4 microplate reader (Bio-tek). Gain was set at 100 and read height at 10.5 mm."

REVIEWER COMMENTS

R1: This manuscript is substantially improved and the authors have addressed many of my comments appropriately.

Minor concern:

The authors' added analysis in Supplementary Figure S2 is helpful and the new data on RUBY and PDC in Supplementary Figure S8 is welcome. However, it is unclear how multiplicative scatter is calculated. I made effort to understand this but could find almost no references or tutorials (nearly all of them instead refer to multiplicative scatter correction). The author's in-caption description in Figure S2 is helpful, but insufficient: "Multiplicative scatter indicates the factor by which the spread of a distribution around the mean is increased by a particular component of the variance, if all other factors are held constant. Multiplicative scatter of 1 indicates that the component contributes no additional variance." Please cite an appropriate source that readers can use to understand this statistical approach or (recommended) include a brief explanation of how this value is calculated.

We thank the reviewer for their thorough and careful reading of our manuscript. The figure caption of Supplementary Fig. 2 has been amended to state: "**A**, multiplicative scatter for each variance component of the mixed effects model fit to 15 independent PCM2:eGFP transient expression experiments (from Fig. 1). Variance components were estimated on the scale to accommodate heteroskedasticity; accordingly, sources of variation are interpreted as multiplicative rather than additive on the original measurement scale. For each component, the estimated standard deviation on the \log_{10} scale ($\sigma_{\log_{10}}$) was calculated, and the back-transformation to the original scale ($10^{\sigma_{\log_{10}}}$), termed multiplicative scatter, was reported. This quantity represents the fold-dispersion around the mean attributable to that component alone, with other components held constant. A value of 1 indicates no additional variability, whereas values >1 indicate increasing multiplicative dispersion (e.g., $\sigma_{\log_{10}} = 0.11 \Rightarrow 10^{0.11} = 1.29$, corresponding to ~29% scatter around the mean). Caterpillar plots show week-specific estimates for between-plant, between-leaf, and residual components."